# A Globally Optimal Portfolio for m-Sparse Sharpe Ratio Maximization

**Yizun Lin**[1]    **Zhao-Rong Lai**[1*]    **Cheng Li**[1]

[1]Department of Mathematics
College of Information Science and Technology
Jinan University, Guangzhou, China
{linyizun,laizhr}@jnu.edu.cn
licheng@stu2020.jnu.edu.cn

## Abstract

The Sharpe ratio is an important and widely-used risk-adjusted return in financial engineering. In modern portfolio management, one may require an $m$-sparse (no more than $m$ active assets) portfolio to save managerial and financial costs. However, few existing methods can optimize the Sharpe ratio with the $m$-sparse constraint, due to the nonconvexity and the complexity of this constraint. We propose to convert the $m$-sparse fractional optimization problem into an equivalent $m$-sparse quadratic programming problem. The semi-algebraic property of the resulting objective function allows us to exploit the Kurdyka-Łojasiewicz property to develop an efficient Proximal Gradient Algorithm (PGA) that leads to a portfolio which achieves the globally optimal $m$-sparse Sharpe ratio under certain conditions. The convergence rates of PGA are also provided. To the best of our knowledge, this is the first proposal that achieves a globally optimal $m$-sparse Sharpe ratio with a theoretically-sound guarantee.

## 1 Introduction

The Sharpe ratio (SR) [33] is an important and widely-used performance metric in finance. Suppose an investing strategy is represented by a portfolio $\boldsymbol{w} \in \mathbb{R}^N$ of $N$ assets from a financial market. $\boldsymbol{\mu} \in \mathbb{R}^N$ and $\boldsymbol{\Sigma} \in \mathbb{R}^{N \times N}$ denote the expected return vector (in excess of the risk-free rate) and its covariance matrix for the $N$ assets, respectively. It can be seen that $\boldsymbol{w}^\top \boldsymbol{\mu}$ and $\sqrt{\boldsymbol{w}^\top \boldsymbol{\Sigma} \boldsymbol{w}}$ represent the expected return and its standard deviation (i.e., risk) for the portfolio $\boldsymbol{w}$. The original definition of SR is given as the follow quotient between return and risk:

$$S_0(\boldsymbol{w}) = \frac{\boldsymbol{w}^\top \boldsymbol{\mu}}{\sqrt{\boldsymbol{w}^\top \boldsymbol{\Sigma} \boldsymbol{w}}}. \tag{1.1}$$

Ever since the proposal of SR, how to maximize it becomes an attractive research topic. Ordinary portfolio optimization methods based on either the mean-variance approach [5, 10] or the exponential growth rate approach [22, 24] can reduce the portfolio risk and increase the portfolio return to some extent [23], and hence improve the SR. On the other hand, direct SR optimization methods are also proposed. Hung et al. [18], Yu and Xu [35] consider the SR as a differentiable function of the portfolio, which can be solved via the augmented Lagrangian method. Pang [29] converts the SR maximization under the self-financing and long-only constraints into a linear complementarity problem, which can be solved via the Parametric Linear Complementarity Technique (PLCT) and the principle pivoting algorithm [12]. Note that PLCT requires $\mu_i > 0$ for at least some asset $i$ in order to be feasible.

---

*Correspondence to: Zhao-Rong Lai

38th Conference on Neural Information Processing Systems (NeurIPS 2024).

In modern portfolio management, it is widely-recognized that the number of selected assets should be restricted to a manageable size, in order to keep simplicity and save time and financial costs. Managerial strategies provide an approach to achieve this objective, such as the revenue driven resource allocation [11], the endowment model [15], and selling stocks after market crashes [27]. However, the managerial approaches still require intensive administration and abundant experience in management and finance. Hence researchers turn to sparsity models for solutions via the computational approaches. In [10], a Sparse and Stable Markowitz Portfolio (SSMP) is proposed by imposing $\ell_1$-regularization on the portfolio. Ao et al. [2] further propose a mean-variance model with an $\ell_1$ constraint based on a maximum-Sharpe-ratio estimation. In [24], the exponential growth rate (EGR) criterion [1, 13, 19, 23] is exploited to develop a Short-term Sparse Portfolio Optimization (SSPO). Furthermore, a Short-term Sparse Portfolio Optimization with $\ell_0$-regularization (SSPO-$\ell_0$) is developed in [28]. In [25], a nonlinear shrinkage of the covariance matrix is proposed to obtain an appropriate size of free parameters. Motivated by this strategy, Lai et al. [22] characterize a sparse structure for covariance estimation to construct a portfolio via the machine learning approach.

The $\ell_1$-regularization and the $\ell_1$ constraint cannot control the exact number of selected assets. One has to tune the sparsity parameter to roughly adjust this number. On the other hand, suppose we want to select no more than $m$ active assets out of $N$ assets to construct a portfolio, then this can be exactly represented by the $m$-sparse (or $\ell_0$) constraint $\|\boldsymbol{w}\|_0 \leqslant m$, where the $\ell_0$ norm $\|\cdot\|_0$ denotes the number of nonzero components of a vector. Although many sparsity models are established for the Markowitz portfolio, few existing methods can optimize the SR (1.1) with the $\ell_0$ constraint, due to the nonconvexity and the complexity of this constraint. In addition to the $\ell_0$ constraint, other realistic constraints should also be imposed to ensure feasibility. For example, the self-financing constraint represents full re-investment and no external loans; the long-only constraint represents no short position. If all these constraints are imposed, the whole model becomes even much more difficult to solve.

To overcome these difficulties, we observe that this optimization is essentially an $m$-sparse fractional optimization that can be transformed into an equivalent $m$-sparse quadratic programming. Then the resulting objective function is semi-algebraic, so that the Kurdyka-Łojasiewicz (KL) property [3] can be exploited to develop an efficient Proximal Gradient Algorithm (PGA) [30]. It converges to a portfolio which achieves the globally optimal $m$-sparse SR under certain conditions. To the best of our knowledge, this is the first proposal that achieves a globally optimal $m$-sparse SR with a theoretically-sound guarantee. Our main contributions can be summarized as follows.

1) We propose to directly maximize the SR with the $\ell_0$ constraint, the self-financing constraint and the long-only constraint on the portfolio. This model aims to obtain a feasible and realistic portfolio that optimizes the SR with exact sparsity.
2) SR maximization is essentially a fractional optimization. We convert this $m$-sparse fractional optimization problem into an equivalent $m$-sparse quadratic programming problem, which reduces the difficulty of solving it.
3) We observe that the resulting objective function is semi-algebraic, thus exploit the KL property to develop an efficient PGA that leads to a globally or at least a locally optimal solution of the $m$-sparse SR maximization model. The convergence rates of PGA are also provided.

Besides the above contributions, our approach also has several advantages: $(i)$ It can be extended to a wide range of optimization problems with semi-algebraic objective functions and constraints. $(ii)$ The actual sparsity is robust to the choice of $m$. $(iii)$ It needs very little parameter tuning. $(iv)$ It does not require any external algorithms or commercial optimizers.

## 2 Related Works and Existing Problems

There are some existing works that indirectly or directly optimize the SR to some extent via the computational approach. We introduce some examples and then analyze some unsolved problems.

### 2.1 Ordinary Portfolio Optimization

An intuitive approach is to directly optimize the portfolio, so that the expected return is maximized and/or the risk is minimized. These methods can be categorized into the mean-variance approach and the exponential growth rate approach [23]. Let $\boldsymbol{R} \in \mathbb{R}^{T \times N}$ be the sample asset return matrix with $T$ trading times and $N$ assets, and $\mathbf{1}_n$ denotes the vector of $n$ ones. Brodie et al. [10] propose

to impose $\ell_1$-regularization on the mean-variance model, forming a Sparse and Stable Markowitz Portfolio (SSMP):

$$\hat{w} = \underset{w \in \mathbb{R}^N}{\mathrm{argmin}} \left\{ \frac{1}{T} \| Rw - \rho \mathbf{1}_T \|_2^2 + \tau \| w \|_1 \right\}, \quad \text{s.t.} \quad w^\top \hat{\mu} = \rho, \ w^\top \mathbf{1}_N = 1, \qquad (2.1)$$

where $\hat{\mu} := \frac{1}{T} R^\top \mathbf{1}_T$ denotes the column vector of sample mean returns, and $\tau \geqslant 0$ is the regularization parameter. $\| \cdot \|_2$ and $\| \cdot \|_1$ denote the $\ell_2$-norm and the $\ell_1$-norm, respectively. The quadratic form $\frac{1}{T} \| Rw - \rho \mathbf{1}_T \|_2^2$ actually computes the mean squared error between the sample portfolio return $r^{(t)} w$ ($r^{(t)}$ is the $t$-th row of $R$) and the given return level $\rho$. Equations $w^\top \hat{\mu} = \rho$ and $w^\top \mathbf{1}_N = 1$ are the expected return constraint and the self-financing constraint, respectively. Model (2.1) can be approximately solved by a surrogate model [10] and the Least Absolute Shrinkage and Selection Operator (Lasso) [34]. Goto and Xu [17] also exploit the Lasso to solve a mean-variance model through sparse hedging restrictions.

Ao et al. [2] propose a maximum-Sharpe-ratio estimated and sparse regression (MAXER) to approach mean-variance efficiency. Assume there are sufficient observations $T > (N + 2)$. MAXER first computes the maximum-Sharpe-ratio estimated regression response $\hat{r}_c$ as follows:

$$\hat{\theta}_s = \hat{\mu}^\top \hat{\Sigma}^{-1} \hat{\mu}, \ \hat{\theta} := \frac{(T - N - 2)\hat{\theta}_s - N}{T}, \ \hat{r}_c := \sigma \frac{1 + \hat{\theta}}{\sqrt{\hat{\theta}}}, \qquad (2.2)$$

where $\hat{\mu}$ and $\hat{\Sigma}$ denote the sample mean and the sample covariance, respectively, $\sigma$ is the risk constraint parameter. Then it adopts the Lasso to obtain the portfolio:

$$\hat{w} = \underset{w \in \mathbb{R}^N}{\mathrm{argmin}} \ \frac{1}{T} \| Rw - \hat{r}_c \mathbf{1}_T \|_2^2, \quad \text{s.t.} \quad \| w \|_1 \leqslant \tau. \qquad (2.3)$$

Instead of the mean-variance approach, our method takes an essentially different objective that directly maximizes the SR in (1.1). Besides, our method does not require $T > (N + 2)$.

Based on the exponential growth rate criterion [1, 13, 19, 23], Lai et al. [24] propose to minimize a kind of negative potential return $w^\top \varphi$ with $\ell_1$-regularization but without any risk term, forming a Short-term Sparse Portfolio Optimization (SSPO) model

$$\hat{w} = \underset{w \in \mathbb{R}^N}{\mathrm{argmin}} \left\{ w^\top \varphi + \tau \| w \|_1 \right\}, \quad \text{s.t.} \quad w^\top \mathbf{1}_N = 1. \qquad (2.4)$$

It develops an unconstrained augmented Lagrangian with the existence of a saddle point that can be solved by the alternating direction method of multipliers (ADMM). Luo et al. [28] further propose the SSPO-$\ell_0$ model

$$\hat{w} = \underset{w \in \Delta}{\mathrm{argmin}} \left\{ w^\top \varphi + \tau \| w \|_0 \right\}, \quad \Delta := \left\{ w \in \mathbb{R}^N \ \middle| \ w \geqslant \mathbf{0}_N \text{ and } w^\top \mathbf{1}_N = 1 \right\}, \qquad (2.5)$$

where $\Delta$ is the $N$-dimensional simplex. This simplex constraint $w \in \Delta$ is the combination of the long-only and the self-financing constraints. Under this constraint, the $\ell_0$-regularization problem (2.5) has a closed-form solution $\tilde{\mathbb{I}}_\varphi^{min} := \left\{ i \in \mathbb{N}_N \ \middle| \ \varphi_i \leqslant \min_{j \in \mathbb{N}_N} \varphi_j + \epsilon \right\}$ with a tolerance $\epsilon \geqslant 0$, where $\mathbb{N}_N := \{1, 2, \ldots, N\}$.

On the other hand, Lai et al. [22] propose a rank-one covariance estimator based on the operator space decomposition, in order to capture the rapidly-changing risk structure in the financial market:

$$X = V_J \Xi U_J^\top, \ D = \Xi^2 - \frac{1}{T} \Xi V_J^\top \mathbf{1}_T \mathbf{1}_T^\top V_J \Xi, \ \zeta_1^* = \left( \frac{\mathrm{tr}(D)}{N(T-1)} \right)^{-\frac{1}{2}} \theta_1, \ \hat{\Sigma}_{RO} := u_1 \zeta_1^* u_1^\top,$$

where $X = R + \mathbf{1}_{T \times N}$ denotes the price relative matrix and $V_J \Xi U_J^\top$ is its singular value decomposition (SVD), $\theta_1$ and $u_1$ are the largest eigenvalue and its eigenvector, respectively.

Although the above portfolio optimization methods may partly improve the SR, they may not be competitive to direct SR optimization. Hence direct SR optimization methods should still be developed and investigated.

## 2.2 Sharpe Ratio Optimization

Pang [29] proposes to optimize the following SR model:

$$\max_{\boldsymbol{w}\in\mathbb{R}^N} S_0(\boldsymbol{w}), \quad \text{s.t.} \quad \boldsymbol{w}\in\Delta,\ \boldsymbol{C}\boldsymbol{w}\leqslant\boldsymbol{d}, \tag{2.6}$$

where $S_0(\boldsymbol{w})$ is defined by (1.1), $\boldsymbol{C}\in\mathbb{R}^{l\times N}$ and $\boldsymbol{d}\in\mathbb{R}^l$ form a linear constraint for $\boldsymbol{w}$. It can be transformed into the following equivalent parametric linear complementarity problem:

$$\begin{cases} \boldsymbol{u} = -\boldsymbol{\mu} + \boldsymbol{\Sigma}\boldsymbol{w} + (\boldsymbol{C}^\top - \mathbf{1}_N \boldsymbol{d}^\top)\boldsymbol{y} \geqslant \mathbf{0}_N, \quad \boldsymbol{w}\geqslant\mathbf{0}_N, \\ \boldsymbol{v} = -(\boldsymbol{C} - \boldsymbol{d}\mathbf{1}_N^\top)\boldsymbol{w} \geqslant \mathbf{0}_l, \quad \boldsymbol{y}\geqslant\mathbf{0}_l, \\ \boldsymbol{u}^\top\boldsymbol{w} = \boldsymbol{v}^\top\boldsymbol{y} = 0. \end{cases} \tag{2.7}$$

Problem (2.7) can be efficiently solved by the principle pivoting algorithm [12], but it requires $\mu_i > 0$ for at least some asset $i$ in order to be feasible. Moreover, if we aim to construct an $m$-sparse $\boldsymbol{w}$, this approach becomes invalid. In Section 3, we will convert the $m$-sparse SR optimization into an equivalent $m$-sparse quadratic programming, and the latter is still a nonconvex optimization. We further elaborate a proximal gradient algorithm to obtain a globally or locally optimal SR.

Another viable approach is to consider the SR as a function of the portfolio and directly optimize it under some realistic constraints. Hung et al. [18] propose the following IPSRM-D model to optimize the SR:

$$\max_{\boldsymbol{w}\in\Delta}\left\{\mathscr{S}(\boldsymbol{w}) := \frac{\boldsymbol{w}^\top\boldsymbol{\mu} + \kappa_1\boldsymbol{w}^\top\boldsymbol{U}\boldsymbol{w}}{\boldsymbol{w}^\top\boldsymbol{D}\boldsymbol{w}} + \kappa_2\boldsymbol{w}^\top(\mathbf{1}_N - \boldsymbol{w})\right\}, \tag{2.8}$$

where $\boldsymbol{U}\in\mathbb{R}^{N\times N}$ and $\boldsymbol{D}\in\mathbb{R}^{N\times N}$ are upside and downside risk matrices, respectively, $\boldsymbol{w}^\top(\mathbf{1}_N - \boldsymbol{w})$ is a diversification term. $\kappa_1$ and $\kappa_2$ are hyperparameters that control the strength of upside risk and diversification, respectively. Interested readers can further refer to [35] for some practical estimators for $\boldsymbol{\mu}, \boldsymbol{U}$ and $\boldsymbol{D}$.

However, $\mathscr{S}(\boldsymbol{w})$ in model (2.8) is different from the original SR $S_0(\boldsymbol{w})$ (1.1) in several significant parts. First, $\mathscr{S}(\boldsymbol{w})$ uses second-order moments $\boldsymbol{w}^\top\boldsymbol{U}\boldsymbol{w}$ and $\boldsymbol{w}^\top\boldsymbol{D}\boldsymbol{w}$ as risk metrics, but $S_0(\boldsymbol{w})$ uses the first-order moment $\sqrt{\boldsymbol{w}^\top\boldsymbol{\Sigma}\boldsymbol{w}}$ instead. In general, a first-order moment is more appropriate because the expected return $\boldsymbol{w}^\top\boldsymbol{\mu}$ should remain in the same order of magnitude as $\sqrt{\boldsymbol{w}^\top\boldsymbol{\Sigma}\boldsymbol{w}}$. Second, the numerator of $S_0(\boldsymbol{w})$ does not contain any risk term, while the numerator of $\mathscr{S}(\boldsymbol{w})$ contains $\boldsymbol{w}^\top\boldsymbol{U}\boldsymbol{w}$. This may change the meaning of SR as an equilibrium point in the efficient frontier based on the CAPM theory [32]. These facts may affect the performance of SR optimization.

Another problem is the lack of effective solving algorithms that could really maximize the SR under constraints. A conventional way is to adopt gradient methods, since $\mathscr{S}(\boldsymbol{w})$ is a differentiable function when $\boldsymbol{w}\neq\mathbf{0}_N$. Hung et al. [18], Yu and Xu [35] propose to adopt the augmented Lagrangian method to optimize (2.8). Though they do not specify which form of Lagrangian models is used, we give the following one without loss of generality:

$$\mathscr{L}(\boldsymbol{w},\boldsymbol{\lambda}) := \mathscr{S}(\boldsymbol{w}) + \frac{\varrho}{2}(\boldsymbol{w}^\top\mathbf{1}_N - 1)^2 + \boldsymbol{\lambda}^\top\boldsymbol{w}, \tag{2.9}$$

where $\frac{\varrho}{2}(\boldsymbol{w}^\top\mathbf{1}_N - 1)^2$ with hyperparameter $\varrho\leqslant 0$ is a regularization term for the self-financing constraint, and $\boldsymbol{\lambda}\in\mathbb{R}_+^N$ is the dual variable with respect to (w.r.t.) $\boldsymbol{w}$ for the long-only constraint $\boldsymbol{w}\geqslant\mathbf{0}_N$. $\mathbb{R}_+^N$ denotes the set of all $N$-dimensional nonnegative vectors. The update scheme is

$$\begin{cases} \boldsymbol{w}^{(k+1)} = \boldsymbol{w}^{(k)} + \eta_1\nabla_{\boldsymbol{w}}\mathscr{L}(\boldsymbol{w}^{(k)},\boldsymbol{\lambda}^{(k)}), \\ \boldsymbol{\lambda}^{(k+1)} = \boldsymbol{\lambda}^{(k)} - \eta_2\nabla_{\boldsymbol{\lambda}}\mathscr{L}(\boldsymbol{w}^{(k+1)},\boldsymbol{\lambda}^{(k)}), \end{cases} \tag{2.10}$$

where $\eta_1,\eta_2\geqslant 0$ are update step sizes. Note that $\mathscr{S}$ is a nonconvex function w.r.t. $\boldsymbol{w}$, and the augmented Lagrangian method is a surrogate method that approximates model (2.8). Hence (2.10) does not necessarily lead to the maximum SR. Worse still, due to the augmented term $\frac{\varrho}{2}(\boldsymbol{w}^\top\mathbf{1}_N - 1)^2$, (2.10) may not even decrease the objective function $\mathscr{S}$. Moreover, the Lagrangian $\mathscr{L}(\boldsymbol{w}^{(k)},\boldsymbol{\lambda}^{(k)})$ is increased by the $\boldsymbol{w}^{(k)}$ updates but decreased by the $\boldsymbol{\lambda}^{(k)}$ updates, hence (2.10) cannot guarantee convergence to a point $(\boldsymbol{w}^*,\boldsymbol{\lambda}^*)$ without a thorough investigation of the update scheme. To summarize, the augmented Lagrangian method and most existing gradient methods cannot guarantee global or local optimality for model (2.8).

# 3 PGA for m-Sparse Sharpe Ratio Maximization

In this section, we formulate the maximization problem of SR as a nonconvex fractional optimization under constraints. Instead of directly solving the proposed model, we develop an efficient proximal gradient algorithm to solve a simpler surrogate model (subtraction form) that is equivalent to the original constrained fractional optimization model.

## 3.1 m-Sparse Sharpe Ratio Maximization Model

In order to retain the risk premium meaning of SR in finance, we directly maximize the original SR in (1.1) instead of a variant like (2.8). In the perspective of statistical estimation, suppose we have a sample asset return (in excess of the risk-free rate) matrix $\boldsymbol{R} \in \mathbb{R}^{T \times N}$ with $T$ trading times and $N$ assets. Then the original SR (1.1) can be represented by

$$S(\boldsymbol{w}) := \frac{\frac{1}{T}\mathbf{1}_T^\top \boldsymbol{R}\boldsymbol{w}}{\sqrt{\frac{1}{T-1}\|\boldsymbol{R}\boldsymbol{w} - (\frac{1}{T}\mathbf{1}_T^\top \boldsymbol{R}\boldsymbol{w})\mathbf{1}_T\|_2^2 + \epsilon\|\boldsymbol{w}\|_2^2}}, \tag{3.1}$$

where $\epsilon\|\boldsymbol{w}\|_2^2$ is a regularization term for a positive definite $\boldsymbol{Q}_\epsilon$ defined in (3.2). The parameter $\epsilon$ can be an arbitrarily-small positive parameter, whose effect on the risk term can be negligible. To simplify the notation, we define

$$\boldsymbol{p} := \frac{1}{T}\boldsymbol{R}^\top \mathbf{1}_T, \;\; \boldsymbol{Q} := \frac{1}{\sqrt{T-1}}\left(\boldsymbol{R} - \frac{1}{T}\mathbf{1}_{T\times T}\boldsymbol{R}\right) \;\text{ and }\; \boldsymbol{Q}_\epsilon := \boldsymbol{Q}^\top \boldsymbol{Q} + \epsilon\boldsymbol{I}. \tag{3.2}$$

Then the maximization of SR under the $m$-sparse, long-only and self-financing constraints is given by

$$\max_{\substack{\boldsymbol{w}\in\Delta \\ \|\boldsymbol{w}\|_0 \leqslant m}} \;\; S(\boldsymbol{w}) := \frac{\boldsymbol{p}^\top \boldsymbol{w}}{\sqrt{\boldsymbol{w}^\top \boldsymbol{Q}_\epsilon \boldsymbol{w}}}, \tag{3.3}$$

where the simplex $\Delta$ is defined in (2.5). Note that minimizing $-S(\boldsymbol{w})$ under the constraint $\|\boldsymbol{w}\|_0 \leqslant m$ is essentially quite different from minimizing the $\ell_0$-regularization version $-S(\boldsymbol{w}) + \tau\|\boldsymbol{w}\|_0$ with some positive $\tau$. In general, the latter is easier because it incorporates the $\ell_0$ norm into the objective function and enlarges the feasible set by dropping the constraint $\|\boldsymbol{w}\|_0 \leqslant m$. We simply call (3.3) the $m$-Sparse Sharpe Ratio Maximization (mSSRM) model. In fact, to solve the mSSRM model, it suffices to solve the following simpler constrained minimization model

$$\min_{\substack{\boldsymbol{v}\geqslant \mathbf{0}_N \\ \|\boldsymbol{v}\|_0 \leqslant m}} \;\; \left\{\frac{1}{2}\boldsymbol{v}^\top \boldsymbol{Q}_\epsilon \boldsymbol{v} - \boldsymbol{p}^\top \boldsymbol{v}\right\}. \tag{3.4}$$

To see this, we establish the relation between the solutions of these two models in the following theorem, whose proof is provided in Appendix A.1. We define the constraint sets in model (3.3) and (3.4) by

$$\Omega_1 := \{\boldsymbol{w} \in \Delta \mid \|\boldsymbol{w}\|_0 \leqslant m\} \;\text{ and }\; \Omega := \{\boldsymbol{v} \in \mathbb{R}^N \mid \boldsymbol{v} \geqslant \mathbf{0}_N \text{ and } \|\boldsymbol{v}\|_0 \leqslant m\}, \tag{3.5}$$

respectively. It is obvious that $\Omega_1 \subsetneqq \Omega$.

**Theorem 1** *Suppose that there exists some $\tilde{\boldsymbol{w}} \in \Omega_1$ such that $\boldsymbol{p}^\top \tilde{\boldsymbol{w}} > 0$. If $\hat{\boldsymbol{v}}$ is an optimal solution of model (3.3), then $\frac{\boldsymbol{p}^\top \hat{\boldsymbol{v}}}{\hat{\boldsymbol{v}}^\top \boldsymbol{Q}_\epsilon \hat{\boldsymbol{v}}}\hat{\boldsymbol{v}}$ is an optimal solution of model (3.4). Conversely, if $\hat{\boldsymbol{v}}$ is an optimal solution of model (3.4), then $\frac{\hat{\boldsymbol{v}}}{\hat{\boldsymbol{v}}^\top \mathbf{1}_N}$ is an optimal solution of model (3.3).*

Defining the indicator function $\iota_\Omega$ by

$$\iota_\Omega(\boldsymbol{v}) := \begin{cases} 0, & if \; \boldsymbol{v} \in \Omega; \\ +\infty, & otherwise, \end{cases} \tag{3.6}$$

we can rewrite model (3.4) as the following two-term unconstrained minimization model:

$$\min_{\boldsymbol{v}\in\mathbb{R}^N} \{f(\boldsymbol{v}) + \iota_\Omega(\boldsymbol{v})\}, \;\text{ where } f(\boldsymbol{v}) := \frac{1}{2}\boldsymbol{v}^\top \boldsymbol{Q}_\epsilon \boldsymbol{v} - \boldsymbol{p}^\top \boldsymbol{v}. \tag{3.7}$$

We then turn to solving model (3.7) instead of the mSSRM model in (3.3).

## 3.2 Proximal Gradient Algorithm

To develop a proximal gradient algorithm for solving model (3.7), we recall the notion of proximity operator, and then establish the relation between the solution of model (3.7) and the proximity characterization (3.8) in Theorem 2. For a proper function $\psi : \mathbb{R}^n \to \overline{\mathbb{R}}$, its proximity operator at $\boldsymbol{x} \in \mathbb{R}^n$ is defined by $\operatorname{prox}_\psi(\boldsymbol{x}) := \operatorname{argmin}_{\boldsymbol{u} \in \mathbb{R}^n} \left\{ \frac{1}{2} \|\boldsymbol{u} - \boldsymbol{x}\|_2^2 + \psi(\boldsymbol{u}) \right\}$. We remark that for function $\psi$ that is nonconvex, $\operatorname{prox}_\psi(\boldsymbol{x})$ may not be unique. Throughout this paper, the formula $\boldsymbol{h} = \operatorname{prox}_\psi(\boldsymbol{x})$ represents $\boldsymbol{h} \in \operatorname{prox}_\psi(\boldsymbol{x})$. For $\boldsymbol{v}^* \in \mathbb{R}^N$ and $\delta > 0$, we denote by $B(\boldsymbol{v}^*; \delta)$ the neighborhood of $\boldsymbol{v}^*$ with radius $\delta$. If there exists some $\delta > 0$ such that $f(\boldsymbol{v}^*) \leqslant f(\boldsymbol{v})$ holds for all $\boldsymbol{v} \in \Omega \cap B(\boldsymbol{v}^*; \delta)$, then we say that $\boldsymbol{v}^*$ is a locally optimal solution of model (3.7).

**Theorem 2** *Let function $\iota_\Omega$ and $f$ be defined by (3.6) and (3.7), respectively. If $\boldsymbol{v}^*$ is a globally optimal solution of model (3.7), then for any $\alpha \in \left( 0, \frac{1}{\|\boldsymbol{Q}_\epsilon\|_2} \right]$,*

$$\boldsymbol{v}^* = \operatorname{prox}_{\iota_\Omega} \left( \boldsymbol{v}^* - \alpha \nabla f(\boldsymbol{v}^*) \right). \tag{3.8}$$

*Conversely, we have the following two statements:*

*(i) If $\alpha \geqslant \frac{1}{\epsilon}$ and (3.8) holds, then $\boldsymbol{v}^*$ is a globally optimal solution of model (3.7).*
*(ii) For any $\alpha > 0$, if (3.8) holds, then $\boldsymbol{v}^*$ is a locally optimal solution of model (3.7).*

The proof of Theorem 2 is provided in Appendix A.2. Based on this theorem, the Proximal Gradient Algorithm (PGA) for solving model (3.7) can be given by the following iterative scheme:

$$\boldsymbol{v}^{(k+1)} = \operatorname{prox}_{\iota_\Omega} \left( \boldsymbol{v}^{(k)} - \alpha \nabla f(\boldsymbol{v}^{(k)}) \right), \quad \text{where } k \in \mathbb{N}, \ \alpha > 0, \ \boldsymbol{v}^{(0)} \in \Omega. \tag{3.9}$$

We then compute the closed form of $\operatorname{prox}_{\iota_\Omega}$. For a vector $\boldsymbol{v} \in \mathbb{R}^N$, we denote by $m_{\boldsymbol{v}}$ and $J_{\text{pos}}^{\boldsymbol{v}}$ the number of positive components and the index set of positive components of $\boldsymbol{v}$. If $m_{\boldsymbol{v}} \geqslant m$, then we denote by $J_{m\text{-pos}}^{\boldsymbol{v}}$ an index set of the $m$-largest positive components of $\boldsymbol{v}$. Specifically, by letting $\{v_{j_i}\}_{i \in \mathbb{N}_N}$ be an rearrangement of $\{v_j\}_{j \in \mathbb{N}_N}$ such that $v_{j_1} \geqslant v_{j_2} \geqslant \cdots \geqslant v_{j_N}$, then $J_{m\text{-pos}}^{\boldsymbol{v}} := \{j_1, j_2, \ldots, j_m\}$. Throughout this paper, for a given vector $\boldsymbol{v} \in \mathbb{R}^N$, we shall always compute $\operatorname{prox}_{\iota_\Omega}(\boldsymbol{v})$ according to the following proposition. Its proof is given in Appendix A.3.

**Proposition 3** *Let $\iota_\Omega$ be defined by (3.6), $\boldsymbol{v} \in \mathbb{R}^N$, and define the index set $J^{\boldsymbol{v}}$ by*

$$J^{\boldsymbol{v}} = \begin{cases} J_{m\text{-pos}}^{\boldsymbol{v}}, & \text{if } m_{\boldsymbol{v}} > m; \\ J_{pos}^{\boldsymbol{v}}, & \text{if } m_{\boldsymbol{v}} \leqslant m. \end{cases}$$

*Then the vector $\boldsymbol{h}$ given by $h_j = \begin{cases} v_j, & \text{if } j \in J^{\boldsymbol{v}}; \\ 0, & \text{if } j \in \mathbb{N}_N \backslash J^{\boldsymbol{v}} \end{cases}$ satisfies that $\boldsymbol{h} \in \operatorname{prox}_{\iota_\Omega}(\boldsymbol{v})$.*

## 3.3 Convergence Analysis of PGA

In this subsection, we delve into the convergence analysis of PGA. We aim to demonstrate that PGA possesses the capability to converge to a globally optimal solution of model (3.4). The limit point obtained by PGA can also yield a globally optimal solution of the original model (3.3), under certain conditions. We also demonstrate the convergence rates of PGA.

Firstly, we introduce a proposition that illustrates the convergence and monotonic decreasing behavior of the objective function values for the iterative sequence, as well as the vanishing gap between consecutive iterates. The proof of this proposition is provided in Appendix A.4.

**Proposition 4** *Let function $\iota_\Omega$ and $f$ be defined by (3.6) and (3.7), respectively, and let $F := f + \iota_\Omega$. If $\alpha \in \left( 0, \frac{1}{\|\boldsymbol{Q}_\epsilon\|_2} \right)$, then for arbitrary initial vector $\boldsymbol{v}^{(0)} \in \mathbb{R}^N$, the sequence $\{\boldsymbol{v}^{(k)}\}_{k \in \mathbb{N}}$ generated by PGA satisfies the following properties:*

*(i) $\boldsymbol{v}^{(k)} \in \Omega$, for all $k \in \mathbb{N}$;*
*(ii) $F(\boldsymbol{v}^{(k+1)}) + a\|\boldsymbol{v}^{(k+1)} - \boldsymbol{v}^{(k)}\|_2^2 \leqslant F(\boldsymbol{v}^{(k)})$ for all $k \in \mathbb{N}$, where $a := \frac{1}{2} \left( \frac{1}{\alpha} - \|\boldsymbol{Q}_\epsilon\|_2 \right) > 0$;*
*(iii) $\lim_{k \to \infty} F(\boldsymbol{v}^{(k)})$ exists;*
*(iv) $\lim_{k \to \infty} \|\boldsymbol{v}^{(k+1)} - \boldsymbol{v}^{(k)}\|_2 = 0$.*

Though we have established the convergence of $\{F(\boldsymbol{v}^{(k)})\}_{k\in\mathbb{N}}$ and the vanishing gap between consecutive iterates, further efforts are necessary to rigorously confirm the convergence of the iterative sequence $\{\boldsymbol{v}^{(k)}\}_{k\in\mathbb{N}}$. We demonstrate the convergence of $\{\boldsymbol{v}^{(k)}\}_{k\in\mathbb{N}}$ to a local minimizer of function $F$ and the corresponding convergence rates in the following theorem. In order to maintain consistency with the original SR maximization model (as outlined in Theorem 1), we further define sequence $\{\boldsymbol{w}^{(k)}\}_{k\in\mathbb{N}}$ based on $\{\boldsymbol{v}^{(k)}\}_{k\in\mathbb{N}}$ and conduct an analysis of the convergence rate of $\{\boldsymbol{w}^{(k)}\}_{k\in\mathbb{N}}$. To prove the theorem, we need to introduce the notions of subdifferential, semi-algebraic function and Kurdyka-Łojasiewicz property, along with several technical lemmas. Detailed proofs and relevant content can be found in Appendix A.5.

**Theorem 5** *Suppose that there exists some $\tilde{\boldsymbol{w}} \in \Omega$ such that $\boldsymbol{p}^\top \tilde{\boldsymbol{w}} > 0$. For arbitrary initial vector $\boldsymbol{v}^{(0)} \in \mathbb{R}^N$, let $\{\boldsymbol{v}^{(k)}\}_{k\in\mathbb{N}}$ be generated by PGA, and let $\{\boldsymbol{w}^{(k)}\}_{k\in\mathbb{N}}$ be defined by*

$$\boldsymbol{w}^{(k)} := \begin{cases} \frac{\boldsymbol{v}^{(k)}}{(\boldsymbol{v}^{(k)})^\top \mathbf{1}_N}, & \text{if } (\boldsymbol{v}^{(k)})^\top \mathbf{1}_N \neq 0; \\ \mathbf{0}_N, & \text{otherwise}. \end{cases}$$

*If $\alpha \in \left(0, \frac{1}{\|\boldsymbol{Q}_\epsilon\|_2}\right)$, then the following statements hold:*

(i) *$\{\boldsymbol{v}^{(k)}\}_{k\in\mathbb{N}}$ converge to a locally optimal solution $\boldsymbol{v}^*$ of model (3.4) with convergence rates $\|\boldsymbol{v}^{(k)} - \boldsymbol{v}^*\|_2 = O(1/\sqrt{k})$ and $|f(\boldsymbol{v}^{(k)}) - f(\boldsymbol{v}^*)| = O(1/k)$.*

(ii) *The limit point $\boldsymbol{v}^*$ of $\{\boldsymbol{v}^{(k)}\}_{k\in\mathbb{N}}$ satisfies that $\boldsymbol{v}^* \geqslant \mathbf{0}_N$ and $\boldsymbol{v}^* \neq \mathbf{0}_N$.*

(iii) *$\{\boldsymbol{w}^{(k)}\}_{k\in\mathbb{N}}$ converge to $\boldsymbol{w}^* := \frac{\boldsymbol{v}^*}{(\boldsymbol{v}^*)^\top \mathbf{1}_N}$ with convergence rates $\|\boldsymbol{w}^{(k)} - \boldsymbol{w}^*\|_2 = O(1/\sqrt{k})$ and $|S(\boldsymbol{w}^{(k)}) - S(\boldsymbol{w}^*)| = O(1/\sqrt{k})$, where $S$ is defined in (3.3).*

In the remainder of this section, we always let $\boldsymbol{v}^* \in \Omega$ be the locally optimal solution of model (3.7) that sequence $\{\boldsymbol{v}^{(k)}\}_{k\in\mathbb{N}}$ converges to. We recall that $m_{\boldsymbol{v}^*}$ and $J_{\text{pos}}^{\boldsymbol{v}^*}$ denote the number of positive components and the index set of positive components of $\boldsymbol{v}^*$, respectively. Suppose that there exists some $\tilde{\boldsymbol{w}} \in \Omega$ such that $\boldsymbol{p}^\top \tilde{\boldsymbol{w}} > 0$. Then item $(ii)$ in Theorem 5 together with $\boldsymbol{v}^* \in \Omega$ yields that $1 \leqslant m_{\boldsymbol{v}^*} \leqslant m$. In fact, $\boldsymbol{v}^*$ is also the globally optimal solution of the convex model

$$\min_{\boldsymbol{v}\in\hat{\Omega}} \left\{ \frac{1}{2}\boldsymbol{v}^\top \boldsymbol{Q}_\epsilon \boldsymbol{v} - \boldsymbol{p}^\top \boldsymbol{v} \right\}, \quad \text{where } \hat{\Omega} := \{\boldsymbol{v} \in \mathbb{R}^N \mid \boldsymbol{v} \geqslant \mathbf{0}_N \text{ and } v_j = 0 \text{ for all } j \in \mathbb{N}_N \setminus J_{\text{pos}}^{\boldsymbol{v}^*}\}. \quad (3.10)$$

Certainly, $\hat{\Omega} \neq \varnothing$ due to the condition $m_{\boldsymbol{v}^*} \geqslant 1$. According to the definition of $\hat{\Omega}$, it is straightforward to observe that $\boldsymbol{v}^* \in \hat{\Omega}$. Furthermore, $\hat{\Omega}$ is a closed convex set and $\hat{\Omega} \subset \Omega$. To analyze the relation between $\boldsymbol{v}^*$ and the original $m$-sparse Sharpe ratio maximization model (3.3), we define

$$\hat{\Omega}_1 := \{\boldsymbol{v} \in \Delta \mid v_j = 0 \text{ for all } j \in \mathbb{N}_N \setminus J_{\text{pos}}^{\boldsymbol{v}^*}\}, \quad (3.11)$$

where $\Delta$ is given by (2.5). It is easy to see that $\hat{\Omega}_1 \subset \Omega_1$, where $\Omega_1$ defined in (3.5) is the constraint set of model (3.3). We then have the following theorem, whose proof is provided in Appendix A.6.

**Theorem 6** *Suppose that there exists some $\tilde{\boldsymbol{w}} \in \hat{\Omega}$ such that $\boldsymbol{p}^\top \tilde{\boldsymbol{w}} > 0$, where $\hat{\Omega}$ is defined in (3.10), and let $\boldsymbol{w}^* := \frac{\boldsymbol{v}^*}{(\boldsymbol{v}^*)^\top \mathbf{1}_N}$. Then the following statements hold:*

(i) *$\boldsymbol{v}^*$ is the unique globally optimal solution of model (3.10).*

(ii) *$\boldsymbol{w}^*$ is a globally optimal solution of model $\max_{\boldsymbol{w}\in\hat{\Omega}_1} S(\boldsymbol{w})$.*

(iii) *If $m_{\boldsymbol{v}^*} = m$, then $\boldsymbol{w}^*$ is a locally optimal solution of model (3.3).*

Item $(iii)$ in Theorem 6 demonstrates that the limit point of the sequence obtained by PGA can yield a locally optimal solution of model (3.3). In fact, according to item $(i)$ in Theorem 2, we have the following theorem that provides sufficient conditions for obtaining a globally optimal solution of model (3.3), whose proof is provided in Appendix A.7.

**Theorem 7** *Suppose that there exists some $\tilde{\boldsymbol{w}} \in \Omega_1$ such that $\boldsymbol{p}^\top \tilde{\boldsymbol{w}} > 0$, and let $\boldsymbol{w}^* := \frac{\boldsymbol{v}^*}{(\boldsymbol{v}^*)^\top \mathbf{1}_N}$. If one of the following two conditions holds:*

(i) *$m_{\boldsymbol{v}^*} < m$;*

(ii) $m_{\boldsymbol{v}^*} = m$ and $\nabla_i f(\boldsymbol{v}^*) > -\epsilon \cdot \min\{v_i^* | i \in \operatorname{supp}(\boldsymbol{v}^*)\}$ for all $i \in \mathbb{N}_N \backslash \operatorname{supp}(\boldsymbol{v}^*)$,

then $\boldsymbol{w}^*$ is a globally optimal solution of model (3.3).

Combining Theorem 7 and item $(iii)$ in Theorem 6, we see that the proposed method can obtain a globally optimal solution of model (3.3) when $m_{\boldsymbol{v}^*} < m$. Even if this condition does not hold, we can obtain at least a locally optimal solution. To test validation of PGA's global optimality, we conduct a set of simulation experiments, whose details are presented in Appendix A.8. The codes for the simulation experiments are accessible via the link: `https://github.com/linyizun2024/mSSRM/tree/main/Codes_for_Simulation`.

We call the existence of $\tilde{\boldsymbol{w}} \in \hat{\Omega}$ such that $\boldsymbol{p}^\top \tilde{\boldsymbol{w}} > 0$ in Theorem 6 the Existence of Positive Expected Return (EPER) condition. Although the EPER condition is required to guarantee the convergence of $\boldsymbol{w}^*$ to a locally optimal solution of the original model (3.3), the proposed method is still of high practical significance in the case that the EPER condition does not hold. From the proofs of item $(i)$ in Theorem 5 and item $(i)$ in Theorem 6, we see that even if the EPER condition does not hold, the sequence generated by PGA still converges to a locally optimal solution $\boldsymbol{v}^*$ of model (3.4), which is also the globally optimal solution of model (3.10). In these two models, the objective function $\frac{1}{2}\boldsymbol{v}^\top \boldsymbol{Q}_\epsilon \boldsymbol{v} - \boldsymbol{p}^\top \boldsymbol{v}$ (subtraction form) w.r.t. $\boldsymbol{v}$ represents risk minus expected return, whose minimization gives smaller risk and less loss in revenue, even if the expected return is not positive. For the case that the expected return is not positive, compared with the failure of Sharpe ratio of fractional form, the globally or locally optimal solution of the subtraction form seems to have more realistic significance. We recall from item $(ii)$ in Theorem 5 that $\boldsymbol{v}^*$ may be equal to $\boldsymbol{0}_N$ if the EPER condition does not hold. In this case, we shall set $\boldsymbol{w}^* = \boldsymbol{0}_N$ and keep all the wealth in the risk-free asset to avoid loss in revenue. To close this section, we summarize the whole $m$-sparse Sharpe ratio maximization method, which we abbreviate to mSSRM-PGA, in Appendix A.9.

## 4 Experimental results

Extensive experiments with real-world financial data sets are conducted to evaluate the performance of the proposed mSSRM-PGA. Moreover, we also consider one baseline method: 1/N [14], as well as 9 state-of-the-art methods: IPSRM-D [18], PLCT [29], SSMP [10], MAXER [2], SSPO [24], SPOLC [22], S1, S2 and S3 [28], as competitors in the experiments. We use 6 real-world monthly benchmark data sets: FF25, FF25EU, FF32, FF49, FF100 and FF100MEINV to compare different methods. These data sets are collected from the baseline and commonly-used Kenneth R. French's Real-world Data Library[2]. Details regarding these competitors and data sets are given in Appendix A.10. As for mSSRM-PGA, we examine three levels of sparsity $m = 10$, $m = 15$, $m = 20$ and set $\epsilon = 10^{-3}$. The setting of other parameters are presented in Appendix A.9. The codes of mSSRM-PGA are accessible via the link: `https://github.com/linyizun2024/mSSRM/tree/main/Codes_for_Experiments_in_Paper`.

### 4.1 Results for Sharpe ratios

We adopt the moving-window trading framework in [23] to imitate real-world portfolio management. For each method, we use the asset returns $\{\boldsymbol{r}_{(t)}\}_{t=1}^T$ or the price relatives $\{\mathbf{x}_{(t)} := \boldsymbol{r}_{(t)} + \mathbf{1}_N\}_{t=1}^T$ in the time window $t = [1 : T]$ to update the portfolio $\hat{\boldsymbol{w}}_{(T+1)}$ for the next trading time. On the $(T + 1)$-th time, we compute the portfolio return by $\hat{r}_{(T+1),\hat{\boldsymbol{w}}} = \mathbf{x}_{(T+1)}^\top \hat{\boldsymbol{w}}_{(T+1)} - 1$ and then turn to the next round where the time window moves to $t = [2 : (T + 1)]$ and a new portfolio $\hat{\boldsymbol{w}}_{(T+2)}$ is computed. This procedure is repeated till the last trading time $\mathscr{T}$, which yields a return sequence $\{\hat{r}_{(t),\hat{\boldsymbol{w}}}\}_{t=1}^{\mathscr{T}}$. This sequence can be used to compute the test SR:

$$\widehat{SR} = \frac{(\sum_{t=T+1}^{\mathscr{T}} \hat{r}_{(t),\hat{\boldsymbol{w}}})/(\mathscr{T} - T)}{\sqrt{(\sum_{s=T+1}^{\mathscr{T}}(\hat{r}_{(s),\hat{\boldsymbol{w}}} - (\sum_{t=T+1}^{\mathscr{T}} \hat{r}_{(t),\hat{\boldsymbol{w}}})/(\mathscr{T} - T))^2)/(\mathscr{T} - T - 1)}}.$$

The 1/N strategy does not involve the time window size $T$. For all other methods, we examine two conventional settings for the time window size in the finance industry [2, 17]: $T = 60$ and $T = 120$.

---

[2] `http://mba.tuck.dartmouth.edu/pages/faculty/ken.french/data_library.html`

Table 1 shows the (monthly) SRs of the 11 compared methods. Because MAXER requires $T > (N+2)$, it is unavailable on FF100 and FF100MEINV when $T = 60$. It is worth noting that the trivial strategy 1/N outperforms most competitors in most situations. The reason is that 1/N diversifies the risk over all the assets, which is also an effective risk control approach [14]. However, mSSRM-PGA outperforms all the competitors including 1/N on all the 6 data sets when $T = 60$ and on 5 data sets when $T = 120$. For example, its SR is more than 70% higher than that of 1/N on FF25EU whether $T = 60$ or $T = 120$. Hence mSSRM-PGA achieves competitive SRs with sparse portfolios, which saves much managerial cost.

Table 1: Sharpe ratios of different portfolio optimization methods on 6 benchmark data sets.

| Strategy | FF25 | FF25EU | FF32 | FF49 | FF100 | FF100MEINV | FF25 | FF25EU | FF32 | FF49 | FF100 | FF100MEINV |
|---|---|---|---|---|---|---|---|---|---|---|---|---|
| | | | | $T = 60$ | | | | | | $T = 120$ | | |
| 1/N | 0.2276 | 0.1574 | 0.2234 | 0.2057 | 0.2087 | 0.2151 | 0.2276 | 0.1574 | 0.2234 | **0.2057** | 0.2087 | 0.2151 |
| SPOLC | 0.1452 | 0.0315 | 0.1734 | 0.0752 | 0.0562 | 0.1009 | 0.1545 | 0.0350 | 0.1830 | 0.1291 | 0.0988 | 0.1218 |
| SSPO | 0.1544 | 0.0411 | 0.1181 | 0.0588 | 0.0425 | 0.0872 | 0.1789 | 0.0719 | 0.1557 | 0.0601 | 0.0529 | 0.1109 |
| S1 | 0.1497 | 0.0369 | 0.1169 | 0.0559 | 0.0327 | 0.0879 | 0.1789 | 0.0736 | 0.1525 | 0.0648 | 0.0467 | 0.0999 |
| S2 | 0.1382 | 0.0633 | 0.1225 | 0.0573 | 0.0456 | 0.1034 | 0.1578 | 0.0725 | 0.1438 | 0.0605 | 0.0602 | 0.1203 |
| S3 | 0.1428 | 0.0607 | 0.1238 | 0.0570 | 0.0469 | 0.1100 | 0.1609 | 0.0709 | 0.1463 | 0.0617 | 0.0603 | 0.1215 |
| SSMP | 0.1934 | 0.1596 | 0.1535 | 0.1658 | 0.0883 | 0.1448 | 0.1920 | 0.0849 | 0.1512 | 0.1581 | 0.0573 | 0.1495 |
| MAXER | 0.1825 | 0.2229 | 0.1625 | 0.1581 | N/A | N/A | 0.1921 | 0.2379 | 0.1465 | 0.1433 | 0.1351 | 0.1479 |
| IPSRM-D | 0.2239 | 0.1994 | 0.1952 | 0.1436 | 0.1766 | 0.1662 | 0.2439 | 0.2358 | 0.2240 | 0.1410 | 0.2012 | 0.1712 |
| PLCT | 0.2475 | 0.2708 | 0.2600 | 0.2119 | 0.2270 | 0.2220 | 0.2468 | 0.2796 | 0.2577 | 0.2025 | 0.2369 | 0.2279 |
| **mSSRM-PGA(m=10)** | **0.2481** | **0.2712** | **0.2612** | **0.2151** | **0.2290** | 0.2217 | **0.2472** | **0.2796** | **0.2592** | 0.2041 | **0.2391** | 0.2271 |
| **mSSRM-PGA(m=15)** | **0.2481** | **0.2708** | **0.2615** | **0.2135** | **0.2289** | **0.2232** | **0.2474** | **0.2796** | **0.2592** | 0.2040 | **0.2381** | **0.2293** |
| **mSSRM-PGA(m=20)** | **0.2481** | **0.2708** | **0.2615** | **0.2134** | **0.2285** | **0.2234** | **0.2474** | **0.2796** | **0.2592** | 0.2041 | **0.2384** | **0.2292** |

## 4.2 Results for Cumulative Wealths

Ordinary investors are also concerned about how much they gain when using an investing strategy. Without loss of generality, we can set the initial wealth for an investing strategy as $S_{(0)} = 1$, then the final cumulative wealth can be conveniently computed by $S_{(\mathcal{T})} = \prod_{t=1}^{\mathcal{T}} (\hat{r}_{(t),\hat{w}} + 1)$. The results of final cumulative wealths are shown in Table 2. The two competitors 1/N and PLCT perform well in general. Nevertheless, mSSRM-PGA achieves the best final cumulative wealths on 4 out of the 6 data sets. Besides, it outperforms each competitor on at least 5 out of the 6 data sets. For example, mSSRM-PGA is about 20% higher than the second best competitor PLCT on FF49 when $T = 60$ and $m = 10$. On the data sets where mSSRM-PGA is not the best method, it is still the second best method. These results indicate that mSSRM-PGA is an effective strategy for pursuing return gain in a practical perspective.

Table 2: Cumulative wealths of different portfolio optimization methods on 6 benchmark data sets.

| Strategy | FF25 | FF25EU | FF32 | FF49 | FF100 | FF100MEINV | FF25 | FF25EU | FF32 | FF49 | FF100 | FF100MEINV |
|---|---|---|---|---|---|---|---|---|---|---|---|---|
| | | | | $T = 60$ | | | | | | $T = 120$ | | |
| 1/N | 355.98 | 13.05 | 424.42 | 235.48 | 364.87 | **428.70** | 355.98 | 13.05 | 424.42 | **235.48** | 364.87 | 428.70 |
| SPOLC | 57.53 | 0.96 | 169.58 | 5.44 | 2.39 | 14.05 | 70.46 | 1.03 | 259.74 | 100.49 | 16.03 | 36.20 |
| SSPO | 129.35 | 1.22 | 30.20 | 1.33 | 0.89 | 8.98 | 286.51 | 2.67 | 130.21 | 1.61 | 1.74 | 25.62 |
| S1 | 100.76 | 1.08 | 29.47 | 1.09 | 0.54 | 9.25 | 265.82 | 2.78 | 121.47 | 2.23 | 1.27 | 15.57 |
| S2 | 66.24 | 2.17 | 39.27 | 1.39 | 1.15 | 20.45 | 130.31 | 2.73 | 93.13 | 1.89 | 2.67 | 43.66 |
| S3 | 70.88 | 2.01 | 36.88 | 1.28 | 1.20 | 23.73 | 129.90 | 2.61 | 89.51 | 1.92 | 2.62 | 38.70 |
| SSMP | 248.67 | 13.47 | 158.98 | 186.79 | 10.09 | 154.27 | 237.45 | 3.25 | 149.65 | 143.18 | 2.26 | 222.35 |
| MAXER | 173.39 | 47.56 | 200.03 | 142.31 | N/A | N/A | 216.94 | 55.71 | 117.42 | 98.85 | 79.82 | 188.54 |
| IPSRM-D | 398.55 | 37.25 | 243.83 | 69.57 | 240.12 | 146.40 | 567.76 | 77.79 | 507.47 | 50.04 | 457.86 | 188.34 |
| PLCT | 581.41 | **126.04** | 918.62 | 238.27 | 471.44 | 354.70 | 608.65 | **148.19** | 854.83 | 157.50 | 552.41 | 399.48 |
| **mSSRM-PGA (m=10)** | **615.34** | 126.02 | **991.89** | **285.02** | **527.09** | 375.75 | **640.89** | 147.17 | **928.19** | 188.38 | **635.65** | 421.97 |
| **mSSRM-PGA (m=15)** | **614.71** | 125.19 | **996.32** | **262.54** | **522.28** | 383.44 | **643.44** | 147.17 | **927.21** | 172.95 | **597.67** | **435.01** |
| **mSSRM-PGA (m=20)** | **614.70** | 125.19 | **996.23** | **262.06** | **515.50** | 384.65 | **643.44** | 147.17 | **927.16** | 173.27 | **603.05** | **433.15** |

## 4.3 Results for Transaction Costs

Cumulative wealth with transaction cost can also be tested to see how the transaction cost influences the performance of different methods. We adopt the proportional transaction cost model [8, 26, 21]

$$S^{\nu} = S_{(0)} \prod_{t=1}^{\mathcal{T}} \left[ (\hat{\boldsymbol{w}}_{(t)}^{\top} \mathbf{x}_{(t)}) \cdot \left( 1 - \frac{\nu}{2} \sum_{i=1}^{N} |\hat{w}_{(t),i} - \tilde{w}_{(t-1),i}| \right) \right], \quad \tilde{w}_{(t-1),i} = \frac{\hat{w}_{(t-1),i} \mathbf{x}_{(t-1),i}}{\boldsymbol{w}_{(t-1)}^{\top} \mathbf{x}_{(t-1)}},$$

where $\tilde{w}_{(t-1),i}$ is the evolved portfolio weight of the $i$-th asset at the end of the $(t-1)$-th period, and $\nu$ is the bidirectional transaction cost rate. When the cost rate of buying is the same as that

of selling, updating the evolved portfolio $\tilde{w}_{(t-1)}$ as the next portfolio $\hat{w}_{(t)}$ yields a proportional transaction cost of $\frac{\nu}{2} \sum_{i=1}^{N} |\hat{w}_{(t),i} - \tilde{w}_{(t-1),i}|$. Figure 2 in Appendix A.10 shows the final cumulative wealths of different methods as $\nu$ varies from 0 to $0.5\%$ with $T = 60$. mSSRM-PGA outperforms all other competitors on FF25, FF25EU and FF32 for all $\nu \in [0, 0.5\%]$, and on FF100 for $\nu \leqslant 0.45\%$. mSSRM-PGA is the second best method on FF100MEINV, following 1/N. This is because 1/N naturally keeps a small trading volume. Note that a manager for a mutual fund with sufficient trades and capital is able to negotiate for a small enough $\nu$. Thus mSSRM-PGA is applicable to scenarios with a certain level of transaction cost.

## 4.4 Sparsity for mSSRM-PGA

In this subsection, we examine the sparsity of the portfolios $\{\hat{w}_{(t)}\}$ generated by mSSRM-PGA. The sparsity can be measured by the cardinality of the support set of $\hat{w}_{(t)}$: $|\mathrm{supp}(\hat{w}_{(t)})|$. For each data set and each setting of $m$, the mean and the standard deviation (STD) of $\{|\mathrm{supp}(\hat{w}_{(t)})|\}$ are computed to provide a general description, shown in Table 3. It indicates that mSSRM-PGA further increases sparsity compared with the preseted sparsity level $m$. Moreover, mSSRM-PGA keeps stable sparsity w.r.t. the change of $m$. For example, the average sparsity for mSSRM-PGA is about $4.9$ when $T = 60$ (or $4.4$ when $T = 120$) on FF25EU, for all the settings $m = 10, 15, 20$. As the total number of assets $N$ increases, mSSRM-PGA gets more advantageous in sparsity. For example, the average sparsity for mSSRM-PGA is about $8 \sim 11$ on FF100 and FF100MEINV, compared with $N = 100$. It indicates that mSSRM-PGA selects only $8\% \sim 11\%$ of the assets in the whole asset pool, while the widely-used 1/N strategy has to maintain the whole asset pool. Therefore, mSSRM-PGA can save much managerial cost by reducing the proportion of selected assets, while keeping a competitive performance in SR optimization.

Table 3: Sparsity of the portfolios generated by mSSRM-PGA: $|\mathrm{supp}(\hat{w}_{(t)})|$.

| m | | FF25 | FF25EU | FF32 | FF49 | FF100 | FF100MEINV | FF25 | FF25EU | FF32 | FF49 | FF100 | FF100MEINV |
|---|---|------|--------|------|------|-------|------------|------|--------|------|------|-------|------------|
| | | | | | $T = 60$ | | | | | | $T = 120$ | | |
| 10 | Mean | 6.3511 | 4.8342 | 6.4159 | 8.1214 | 8.0097 | 7.9175 | 7.1359 | 4.4560 | 7.0825 | 8.4790 | 8.3706 | 8.8722 |
| | STD | 2.4164 | 2.1763 | 2.3654 | 1.9918 | 2.2473 | 2.3915 | 2.2221 | 1.4645 | 2.4464 | 1.9080 | 2.3343 | 2.0117 |
| 15 | Mean | 6.4746 | 4.9352 | 7.4286 | 9.0000 | 8.9709 | 9.0825 | 7.1637 | 4.4430 | 7.4919 | 9.6003 | 9.8754 | 10.5906 |
| | STD | 3.0451 | 2.3573 | 3.0590 | 3.0713 | 3.3964 | 3.5906 | 2.2995 | 1.4462 | 2.2567 | 3.1731 | 3.6169 | 3.4845 |
| 20 | Mean | 6.4763 | 4.9352 | 7.4692 | 9.0421 | 9.1974 | 9.2994 | 7.1637 | 4.4430 | 7.4919 | 9.6828 | 10.0437 | 10.7621 |
| | STD | 3.0462 | 2.3573 | 3.1734 | 3.1620 | 3.8949 | 4.0125 | 2.2995 | 1.4462 | 2.2567 | 3.3349 | 3.9160 | 3.7460 |

## 5 Concluding Remarks

The Sharpe ratio (SR) is a very important measurement for the performance of returns attributable to risk in finance. On the other hand, modern portfolio management usually restricts the number of selected assets to a relatively small size, in order to save managerial and financial costs. The $m$-sparse ($\ell_0$) constraint is an exact constraint for a sparse portfolio, but it is nonconvex and complex. Thus few existing methods can optimize the SR with the $m$-sparse constraint. In this study, we convert the $m$-sparse fractional optimization problem into an equivalent $m$-sparse quadratic programming problem. Then we develop an efficient, easy-to-implement and mathematically sound proximal gradient algorithm to solve this nonconvex problem. We theoretically prove that this algorithm yields a portfolio that achieves the globally optimal $m$-sparse Sharpe ratio under certain conditions.

We conduct extensive experiments on 6 real-world monthly benchmark data sets built on the Kenneth R. French's widely-used public data library. The numerical results demonstrate that the proposed mSSRM-PGA improves the SR, compared with 9 state-of-the-art portfolio optimization methods including SPOLC, SSPO, S1, S2, S3, SSMP, MAXER, IPSRM-D, PLCT and one baseline method 1/N. For another evaluating metric cumulative wealth, mSSRM-PGA outperforms each competitor on at least 5 out of the 6 data sets. Besides, mSSRM-PGA can withstand a considerable level of transaction cost rate. Sparsity experiments indicate that mSSRM-PGA successfully generates portfolios with stable sparsity, and its advantage increases as the size of the whole asset pool increases. In summary, the proposed mSSRM-PGA is a promising approach in managing portfolios or other financial issues, which is worth further investigations. A limitation of this research lies in its inability to directly apply to fractional optimization models featuring nondifferentiable numerator and denominator. Future work will strive to broaden the theoretical and methodological foundations, ultimately enabling its application to a broader spectrum of fractional optimization models in machine learning.

## Acknowledgements

The authors thank the anonymous reviewers for their constructive comments and valuable suggestions in improving this paper. This work was supported in part by National Natural Science Foundation of China under Grants 12401120 and 62176103, in part by Guangdong Basic and Applied Basic Research Foundation under Grants 2021A1515110541 and 2023B1515120064, in part by the Science and Technology Planning Project of Guangdong under Grant 2023A0505030013, and in part by the Science and Technology Planning Project of Guangzhou under Grants 2024A04J3940, 2024A04J9896, 202206030007, Nansha District: 2023ZD001 and Development District: 2023GH01.

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

# A   Appendix

## A.1   Proof of Theorem 1

To prove Theorem 1, we need the following lemma.

**Lemma 8** *Suppose that there exists some $\tilde{w} \in \Omega_1$ such that $p^\top \tilde{w} > 0$. If $\hat{v}$ is an optimal solution of model (3.4), then $\hat{v} \neq \mathbf{0}_N$ and $p^\top \hat{v} = \hat{v}^\top Q_\epsilon \hat{v} > 0$.*

**Proof.** Since $\hat{v}$ is an optimal solution of model (3.4), we know that $\hat{v} \in \Omega$. Let $w := \frac{p^\top \tilde{w}}{\tilde{w}^\top Q_\epsilon \tilde{w}} \tilde{w}$. The facts $\tilde{w} \in \Omega_1$ and $p^\top \tilde{w} > 0$ imply that $w \in \Omega$. Then it follows that

$$\frac{1}{2} \hat{v}^\top Q_\epsilon \hat{v} - p^\top \hat{v} \leqslant \frac{1}{2} w^\top Q_\epsilon w - p^\top w = -\frac{1}{2} \frac{(p^\top \tilde{w})^2}{\tilde{w}^\top Q_\epsilon \tilde{w}} < 0.$$

Hence $p^\top \hat{v} > \frac{1}{2} \hat{v}^\top Q_\epsilon \hat{v} \geqslant 0$ and $\hat{v} \neq \mathbf{0}_N$. Now by letting $v := \frac{p^\top \hat{v}}{\hat{v}^\top Q_\epsilon \hat{v}} \hat{v}$, then $v \in \Omega$ and

$$\frac{1}{2} \hat{v}^\top Q_\epsilon \hat{v} - p^\top \hat{v} \leqslant \frac{1}{2} v^\top Q_\epsilon v - p^\top v = -\frac{1}{2} \frac{(p^\top \hat{v})^2}{\hat{v}^\top Q_\epsilon \hat{v}}. \tag{A.1}$$

Multiplying both sides of (A.1) by $2\hat{v}^\top Q_\epsilon \hat{v}$ yields

$$\left( p^\top \hat{v} - \hat{v}^\top Q_\epsilon \hat{v} \right)^2 \leqslant 0,$$

which implies that $p^\top \hat{v} = \hat{v}^\top Q_\epsilon \hat{v} > 0$. □

**Proof of Theorem 1.** Let $\hat{v}$ be an optimal solution of model (3.3). Then $\hat{v} \in \Omega_1$ and

$$\frac{p^\top \hat{v}}{\sqrt{\hat{v}^\top Q_\epsilon \hat{v}}} \geqslant \frac{p^\top \tilde{w}}{\sqrt{\tilde{w}^\top Q_\epsilon \tilde{w}}} > 0.$$

Defining $\tilde{v} := \frac{p^\top \hat{v}}{\hat{v}^\top Q_\epsilon \hat{v}} \hat{v}$, we see that $\tilde{v} \in \Omega$ and

$$\frac{1}{2} \tilde{v}^\top Q_\epsilon \tilde{v} - p^\top \tilde{v} = \frac{1}{2} \frac{(p^\top \hat{v})^2}{(\hat{v}^\top Q_\epsilon \hat{v})^2} \hat{v}^\top Q_\epsilon \hat{v} - \frac{(p^\top \hat{v})^2}{\hat{v}^\top Q_\epsilon \hat{v}} = -\frac{1}{2} \frac{(p^\top \hat{v})^2}{\hat{v}^\top Q_\epsilon \hat{v}} < 0. \tag{A.2}$$

For any $u \in \Omega$ such that $\frac{1}{2} u^\top Q_\epsilon u - p^\top u < 0$, we have $p^\top u > 0$, $u \neq \mathbf{0}_N$ and $\tilde{u} := \frac{u}{u^\top \mathbf{1}_N} \in \Omega_1$. Then the fact $\hat{v}$ is an optimal solution of model (3.3) implies that

$$\frac{(p^\top \hat{v})^2}{\hat{v}^\top Q_\epsilon \hat{v}} \geqslant \frac{(p^\top \tilde{u})^2}{\tilde{u}^\top Q_\epsilon \tilde{u}} = \frac{(p^\top u)^2}{u^\top Q_\epsilon u}. \tag{A.3}$$

Note that

$$\frac{1}{2} u^\top Q_\epsilon u - p^\top u + \frac{1}{2} \frac{(p^\top u)^2}{u^\top Q_\epsilon u} = \frac{1}{2 u^\top Q_\epsilon u} (u^\top Q_\epsilon u - p^\top u)^2 \geqslant 0,$$

which combined with (A.3) and (A.2) yields

$$\frac{1}{2} u^\top Q_\epsilon u - p^\top u \geqslant -\frac{1}{2} \frac{(p^\top u)^2}{u^\top Q_\epsilon u} \geqslant -\frac{1}{2} \frac{(p^\top \hat{v})^2}{\hat{v}^\top Q_\epsilon \hat{v}} = \frac{1}{2} \tilde{v}^\top Q_\epsilon \tilde{v} - p^\top \tilde{v}.$$

Therefore, $\tilde{v}$ is an optimal solution of model (3.4).

Conversely, let $\hat{v}$ be an optimal solution of model (3.4). It follows from Lemma 8 that $\hat{v} \neq \mathbf{0}_N$ and $p^\top \hat{v} = \hat{v}^\top Q_\epsilon \hat{v} > 0$. Thus $\hat{v} = \frac{p^\top \hat{v}}{\hat{v}^\top Q_\epsilon \hat{v}} \hat{v}$. For any $v \in \Omega$ such that $p^\top v > 0$, we let $u := \frac{p^\top v}{v^\top Q_\epsilon v} v$. Then $u \in \Omega$ and

$$-\frac{1}{2} \frac{(p^\top \hat{v})^2}{\hat{v}^\top Q_\epsilon \hat{v}} = \frac{1}{2} \hat{v}^\top Q_\epsilon \hat{v} - p^\top \hat{v} \leqslant \frac{1}{2} u^\top Q_\epsilon u - p^\top u = -\frac{1}{2} \frac{(p^\top v)^2}{v^\top Q_\epsilon v}. \tag{A.4}$$

Now we let $\bar{v} := \frac{\hat{v}}{\hat{v}^\top \mathbf{1}_N}$. Then $\bar{v} \in \Omega_1$. Inequality (A.4) yields that

$$\frac{p^\top \bar{v}}{\sqrt{\bar{v}^\top Q_\epsilon \bar{v}}} = \frac{p^\top \hat{v}}{\sqrt{\hat{v}^\top Q_\epsilon \hat{v}}} \geqslant \frac{p^\top v}{\sqrt{v^\top Q_\epsilon v}}.$$

Note that $\Omega_1 \subset \Omega$. Therefore, $\bar{v}$ is an optimal solution of model (3.3). □

## A.2 Proof of Theorem 2

To prove Theorem 2, we first investigate the properties of function $f$ in Proposition 9, and then recall two well-known results as Lemmas 10 and 11. Let $\psi$ be a function from $\mathbb{R}^n$ to $[-\infty, +\infty]$. Then $\psi$ is proper if $-\infty \notin \psi(\mathbb{R}^n)$ and $\{x \in \mathbb{R}^n \,|\, \psi(x) < +\infty\} \neq \varnothing$. Let $\psi : \mathbb{R}^n \to \overline{\mathbb{R}}$ be a proper function. We say that $\psi$ is convex if for any $x, y \in \mathbb{R}^n$ and any $\lambda \in (0, 1)$, $\psi(\lambda x + (1-\lambda)y) \leqslant \lambda\psi(x) + (1-\lambda)\psi(y)$. If there exists $\beta > 0$ such that $\psi - \frac{\beta}{2}\|\cdot\|_2^2$ is convex, then $\psi$ is said to be $\beta$-strongly convex.

**Proposition 9** *Let $f : \mathbb{R}^N \to \mathbb{R}$ be defined in* (3.7). *Then the following hold:*

$(i)$ *$f$ is $\epsilon$-strongly convex on $\mathbb{R}^N$;*

$(ii)$ *$\nabla f$ is $\|\boldsymbol{Q}_\epsilon\|_2$-Lipschitz continuous on $\mathbb{R}^N$.*

***Proof.*** Let $\tilde{f}(\boldsymbol{v}) := f(\boldsymbol{v}) - \frac{\epsilon}{2}\|\boldsymbol{v}\|_2^2 = \frac{1}{2}\boldsymbol{v}^\top \boldsymbol{Q}^\top \boldsymbol{Q} \boldsymbol{v} - \boldsymbol{p}^\top \boldsymbol{v}$, $\boldsymbol{v} \in \mathbb{R}^N$. Since the Hessian matrix $\boldsymbol{Q}^\top \boldsymbol{Q}$ of $\tilde{f}$ is positive semidefinite, we know that $\tilde{f}$ is convex on $\mathbb{R}^N$ (see Proposition B.4 of [7]). Thus item $(i)$ holds. The gradient of $f$ is given by $\nabla f(\boldsymbol{v}) = \boldsymbol{Q}_\epsilon \boldsymbol{v} - \boldsymbol{p}$. For all $\boldsymbol{x}, \boldsymbol{y} \in \mathbb{R}^N$, $\|\nabla f(\boldsymbol{x}) - \nabla f(\boldsymbol{y})\|_2 \leqslant \|\boldsymbol{Q}_\epsilon\|_2\|\boldsymbol{x} - \boldsymbol{y}\|_2$, which implies item $(ii)$. $\qquad\square$

**Lemma 10 (Proposition A.24 of [7])** *Let function $\psi : \mathbb{R}^n \to \mathbb{R}$ be differentiable with an $L$-Lipschitz continuous gradient, where $L > 0$. Then*

$$\psi(\boldsymbol{y}) - \psi(\boldsymbol{x}) \leqslant \langle \nabla\psi(\boldsymbol{x}), \boldsymbol{y} - \boldsymbol{x}\rangle + \frac{L}{2}\|\boldsymbol{y} - \boldsymbol{x}\|_2^2$$

*holds for all $\boldsymbol{x}, \boldsymbol{y} \in \mathbb{R}^n$.*

**Lemma 11 (Exercise 17.5 of [6])** *Let $\psi : \mathbb{R}^n \to \mathbb{R}$ be differentiable and $\beta > 0$. Then $\psi$ is $\beta$-strongly convex if and only if*

$$\psi(\boldsymbol{y}) - \psi(\boldsymbol{x}) \geqslant \langle \nabla\psi(\boldsymbol{x}), \boldsymbol{y} - \boldsymbol{x}\rangle + \frac{\beta}{2}\|\boldsymbol{y} - \boldsymbol{x}\|_2^2$$

*holds for all $\boldsymbol{x}, \boldsymbol{y} \in \mathbb{R}^n$.*

***Proof of Theorem 2.*** We first show that (3.8) holds when $\boldsymbol{v}^*$ is a globally optimal solution of model (3.7). By the definition of proximity operator, (3.8) is equivalent to

$$\boldsymbol{v}^* = \operatorname*{argmin}_{\boldsymbol{u}\in\mathbb{R}^N} \iota_\Omega(\boldsymbol{u}) + \frac{1}{2}\|\boldsymbol{u} - \boldsymbol{v}^* + \alpha\nabla f(\boldsymbol{v}^*)\|_2^2,$$

that is,

$$\iota_\Omega(\boldsymbol{u}) + \frac{1}{2}\|\boldsymbol{u} - \boldsymbol{v}^* + \alpha\nabla f(\boldsymbol{v}^*)\|_2^2 \geqslant \iota_\Omega(\boldsymbol{v}^*) + \frac{1}{2}\|\alpha\nabla f(\boldsymbol{v}^*)\|_2^2, \text{ for all } \boldsymbol{u} \in \mathbb{R}^N.$$

According to the definition of $\iota_\Omega$ in (3.6) and the fact $\boldsymbol{v}^* \in \Omega$, the above inequality can be simply rewritten as

$$\langle \nabla f(\boldsymbol{v}^*), \boldsymbol{u} - \boldsymbol{v}^*\rangle + \frac{1}{2\alpha}\|\boldsymbol{u} - \boldsymbol{v}^*\|_2^2 \geqslant 0, \text{ for all } \boldsymbol{u} \in \Omega. \tag{A.5}$$

To prove (3.8), it suffices to show that (A.5) holds. From Proposition 9, we know that $f$ is $\epsilon$-strongly convex and $\nabla f$ is $\|\boldsymbol{Q}_\epsilon\|$-Lipschitz continuous on $\mathbb{R}^N$. Then Lemma 10 yields that

$$f(\boldsymbol{u}) - f(\boldsymbol{v}^*) \leqslant \langle \nabla f(\boldsymbol{v}^*), \boldsymbol{u} - \boldsymbol{v}^*\rangle + \frac{\|\boldsymbol{Q}_\epsilon\|_2}{2}\|\boldsymbol{u} - \boldsymbol{v}^*\|_2^2, \text{ for all } \boldsymbol{u} \in \Omega. \tag{A.6}$$

Since $\boldsymbol{v}^*$ is a globally optimal solution of model (3.7), $f(\boldsymbol{u}) - f(\boldsymbol{v}^*) \geqslant 0$ for all $\boldsymbol{u} \in \Omega$, which together with (A.6) and the fact $\alpha \in \left(0, \frac{1}{\|\boldsymbol{Q}_\epsilon\|_2}\right]$ implies (A.5). This proves that (3.8) holds.

Conversely, if $\alpha \geqslant \frac{1}{\epsilon}$ and (3.8) holds, then we have (A.5). Recall that $f$ is $\epsilon$-strongly convex. It follows from Lemma 11 that

$$f(\boldsymbol{u}) - f(\boldsymbol{v}^*) \geqslant \langle \nabla f(\boldsymbol{v}^*), \boldsymbol{u} - \boldsymbol{v}^*\rangle + \frac{\epsilon}{2}\|\boldsymbol{u} - \boldsymbol{v}^*\|_2^2,$$

which together with the fact $\alpha \geqslant \frac{1}{\epsilon}$ and (A.5) implies that $f(\boldsymbol{u}) - f(\boldsymbol{v}^*) \geqslant 0$ for all $\boldsymbol{u} \in \Omega$. Thus the assertion in item $(i)$ holds.

We then prove item $(ii)$. The fact (3.8) holds implies (A.5). For $\delta > 0$, we define

$$\tilde{\Omega}_\delta := \{\boldsymbol{u} \in B(\boldsymbol{v}^*; \delta) | \langle \nabla f(\boldsymbol{v}^*), \boldsymbol{u} - \boldsymbol{v}^* \rangle = 0\}.$$

Note that when $\boldsymbol{u}$ tends to $\boldsymbol{v}^*$, the quadratic term $\frac{1}{2\alpha} \|\boldsymbol{u} - \boldsymbol{v}^*\|_2^2$ is of higher order infinitesimal than the linear term $|\langle \nabla f(\boldsymbol{v}^*), \boldsymbol{u} - \boldsymbol{v}^* \rangle|$. There must be some $\delta > 0$ such that

$$|\langle \nabla f(\boldsymbol{v}^*), \boldsymbol{u} - \boldsymbol{v}^* \rangle| > \frac{1}{2\alpha} \|\boldsymbol{u} - \boldsymbol{v}^*\|_2^2, \ \ \text{for all } \boldsymbol{u} \in B(\boldsymbol{v}^*; \delta) \backslash \tilde{\Omega}_\delta. \tag{A.7}$$

We then show that

$$\langle \nabla f(\boldsymbol{v}^*), \boldsymbol{u} - \boldsymbol{v}^* \rangle \geqslant 0, \ \ \text{for all } \boldsymbol{u} \in \Big( B(\boldsymbol{v}^*; \delta) \backslash \tilde{\Omega}_\delta \Big) \cap \Omega. \tag{A.8}$$

Otherwise, there exists some $\tilde{\boldsymbol{u}} \in \Big( B(\boldsymbol{v}^*; \delta) \backslash \tilde{\Omega}_\delta \Big) \cap \Omega$ such that $\langle \nabla f(\boldsymbol{v}^*), \tilde{\boldsymbol{u}} - \boldsymbol{v}^* \rangle < 0$. It follows from (A.7) that

$$\langle \nabla f(\boldsymbol{v}^*), \tilde{\boldsymbol{u}} - \boldsymbol{v}^* \rangle + \frac{1}{2\alpha} \|\tilde{\boldsymbol{u}} - \boldsymbol{v}^*\|_2^2 < 0,$$

which contradicts (A.5). Hence (A.8) holds. This together with the definition of $\tilde{\Omega}_\delta$ yields that

$$\langle \nabla f(\boldsymbol{v}^*), \boldsymbol{u} - \boldsymbol{v}^* \rangle \geqslant 0, \ \ \text{for all } \boldsymbol{u} \in B(\boldsymbol{v}^*; \delta) \cap \Omega. \tag{A.9}$$

Recall that $f$ is convex and differentiable on $\mathbb{R}^N$. According to (A.9) and the first order condition for convexity (Proposition B.3 of [7]),

$$f(\boldsymbol{u}) - f(\boldsymbol{v}^*) \geqslant \langle \nabla f(\boldsymbol{v}^*), \boldsymbol{u} - \boldsymbol{v}^* \rangle \geqslant 0, \ \ \text{for all } \boldsymbol{u} \in B(\boldsymbol{v}^*; \delta) \cap \Omega,$$

which implies that $\boldsymbol{v}^*$ is a locally optimal solution of model (3.7). □

### A.3 Proof of Proposition 3

***Proof.*** By the definitions of $\iota_\Omega$ and its proximity operator, we have

$$\mathrm{prox}_{\iota_\Omega}(\boldsymbol{v}) = \underset{\boldsymbol{u} \in \Omega}{\mathrm{argmin}} \|\boldsymbol{u} - \boldsymbol{v}\|_2.$$

To prove that $\boldsymbol{h} \in \mathrm{prox}_{\iota_\Omega}(\boldsymbol{v})$, it is equivalent to show that

$$\|\boldsymbol{h} - \boldsymbol{v}\|_2^2 \leqslant \|\boldsymbol{u} - \boldsymbol{v}\|_2^2, \ \ \text{for all } \boldsymbol{u} \in \Omega. \tag{A.10}$$

For any $\boldsymbol{u} \in \Omega$, there exists an index set $J_{\boldsymbol{u}} \in \mathbb{N}_N$ with $m$ elements such that $u_j = 0$ for all $j \in \mathbb{N}_N \backslash J_{\boldsymbol{u}}$. Let $J_{\mathrm{neg}}^{\boldsymbol{v}}$ be the index set of negative components in $\boldsymbol{v}$ and $J_{\boldsymbol{u}}' := (\mathbb{N}_N \backslash J_{\boldsymbol{u}}) \cup J_{\mathrm{neg}}^{\boldsymbol{v}}$. Since $\boldsymbol{u} \geqslant \boldsymbol{0}_N$, $\|\boldsymbol{u} - \boldsymbol{v}\|_2^2 \geqslant \sum_{j \in J_{\boldsymbol{u}}'} v_j^2$. Let $J_{\boldsymbol{h}}' = \mathbb{N}_N \backslash J^{\boldsymbol{v}}$. Then $J_{\mathrm{neg}}^{\boldsymbol{v}} \subset J_{\boldsymbol{h}}'$ and $\|\boldsymbol{h} - \boldsymbol{v}\|_2^2 = \sum_{j \in J_p'} v_j^2$. If $m_{\boldsymbol{v}} > m$, then $J^{\boldsymbol{v}} = J_{m\text{-pos}}^{\boldsymbol{v}}$. We are easy to see from the definition of $J_{m\text{-pos}}^{\boldsymbol{v}}$ that

$$\sum_{j \in J_{\boldsymbol{u}}'} v_j^2 - \sum_{j \in J_{\boldsymbol{h}}'} v_j^2 = \sum_{j \in \mathbb{N}_N \backslash (J_{\boldsymbol{u}} \cup J_{\mathrm{neg}}^{\boldsymbol{v}})} v_j^2 - \sum_{j \in \mathbb{N}_N \backslash (J_{m\text{-pos}}^{\boldsymbol{v}} \cup J_{\mathrm{neg}}^{\boldsymbol{v}})} v_j^2 \geqslant 0.$$

If $m_{\boldsymbol{v}} \leqslant m$, then $J^{\boldsymbol{v}} = J_{\mathrm{pos}}^{\boldsymbol{v}}$ and $\sum_{j \in J_{\boldsymbol{u}}'} v_j^2 - \sum_{j \in J_{\boldsymbol{h}}'} v_j^2 = \sum_{j \in (\mathbb{N}_N \backslash J_{\boldsymbol{u}}) \cup J_{\mathrm{neg}}^{\boldsymbol{v}}} v_j^2 - \sum_{j \in J_{\mathrm{neg}}^{\boldsymbol{v}}} v_j^2 \geqslant 0$. Now we conclude from the above two cases that

$$\|\boldsymbol{u} - \boldsymbol{v}\|_2^2 - \|\boldsymbol{h} - \boldsymbol{v}\|_2^2 \geqslant \sum_{j \in J_{\boldsymbol{u}}'} v_j^2 - \sum_{j \in J_{\boldsymbol{h}}'} v_j^2 \geqslant 0,$$

that is, (A.10) holds. This completes the proof. □

## A.4 Proof of Proposition 4

**Proof.** Item $(i)$ follows from (3.9) and the definition of $\mathrm{prox}_{\iota_\Omega}$ directly. Then we have that $\iota_\Omega(\boldsymbol{v}^{(k)}) = 0$ for all $k \in \mathbb{N}$. To prove item $(ii)$, it suffices to show that

$$f(\boldsymbol{v}^{(k+1)}) + a\|\boldsymbol{v}^{(k+1)} - \boldsymbol{v}^{(k)}\|_2^2 \leqslant f(\boldsymbol{v}^{(k)}), \quad \text{for all } k \in \mathbb{N}. \tag{A.11}$$

Note that $a = \frac{1}{\alpha} - \|\boldsymbol{Q}_\epsilon\|_2 > 0$, since $\alpha \in \left(0, \frac{1}{\|\boldsymbol{Q}_\epsilon\|_2}\right)$. Let

$$\varphi(\boldsymbol{u}) := \frac{1}{2}\left\|\boldsymbol{u} - \boldsymbol{v}^{(k)} + \alpha \nabla f(\boldsymbol{v}^{(k)})\right\|_2^2 + \iota_\Omega(\boldsymbol{u}), \quad \boldsymbol{u} \in \mathbb{R}^N. \tag{A.12}$$

Then (3.9) implies that $\varphi(\boldsymbol{v}^{(k+1)}) \leq \varphi(\boldsymbol{v}^{(k)})$, that is,

$$\langle \nabla f(\boldsymbol{v}^{(k)}), \boldsymbol{v}^{(k+1)} - \boldsymbol{v}^{(k)} \rangle \leqslant -\frac{1}{2\alpha}\|\boldsymbol{v}^{(k+1)} - \boldsymbol{v}^{(k)}\|_2^2, \quad \text{for all } k \in \mathbb{N}, \tag{A.13}$$

It follows from Lemma 10 that

$$f(\boldsymbol{v}^{(k+1)}) - f(\boldsymbol{v}^{(k)}) \leqslant \langle \nabla f(\boldsymbol{v}^{(k)}), \boldsymbol{v}^{(k+1)} - \boldsymbol{v}^{(k)} \rangle + \frac{\|\boldsymbol{Q}_\epsilon\|_2}{2}\|\boldsymbol{v}^{(k+1)} - \boldsymbol{v}^{(k)}\|_2^2. \tag{A.14}$$

Combining (A.13) and (A.14) yields (A.11). Thus item $(ii)$ holds. Now that $F$ is monotonically decreasing, according to the monotone convergence theorem, to prove item $(iii)$, it suffices to show that function $F$ is bounded below on $\Omega$. Solving $\nabla f(\boldsymbol{v}^*) = 0$ gives $\boldsymbol{v}^* = \boldsymbol{Q}_\epsilon^{-1}\boldsymbol{p}$. Since $f$ is convex and differentiable on $\mathbb{R}^N$, $f$ attains the minimum value at $\boldsymbol{Q}_\epsilon^{-1}\boldsymbol{p}$ on $\mathbb{R}^N$. Hence $f(\boldsymbol{v}) \geqslant f\left(\boldsymbol{Q}_\epsilon^{-1}\boldsymbol{p}\right) = -\frac{1}{2}\boldsymbol{p}^\top\boldsymbol{Q}_\epsilon^{-1}\boldsymbol{p}$ for all $\boldsymbol{v} \in \mathbb{R}^N$, which implies that $F(\boldsymbol{v}) \geqslant -\frac{1}{2}\boldsymbol{p}^\top\boldsymbol{Q}_\epsilon^{-1}\boldsymbol{p}$ for all $\boldsymbol{v} \in \Omega$. Therefore, item $(iii)$ holds. Now taking the limit on both sides of the inequality in item $(ii)$ yields item $(iv)$ immediately. This completes the proof. □

## A.5 Proof of Theorem 5

In order to prove Theorem 5, it is necessary to review several definitions and establish several preliminary results. First, We recall the notions of subdifferentials and critical point. The lower limit of function $\psi$ at $\boldsymbol{x}$ and the domain of $\psi$ are defined by

$$\liminf_{\boldsymbol{y} \to \boldsymbol{x}} \psi(\boldsymbol{y}) := \lim_{\delta \to 0^+}\left(\inf_{\boldsymbol{y} \in B(\boldsymbol{x};\delta)} \psi(\boldsymbol{y})\right) \tag{A.15}$$

and

$$\mathrm{dom}\,\psi := \{\boldsymbol{x} \in \mathbb{R}^n | \psi(\boldsymbol{x}) < +\infty\},$$

respectively. We say that $\psi$ is lower semicontinuous at $\boldsymbol{x} \in \mathbb{R}^n$ if $\psi(\boldsymbol{x}) \leqslant \liminf_{\boldsymbol{u} \to \boldsymbol{x}} \psi(\boldsymbol{u})$. If $\psi$ is lower semicontinuous at every $\boldsymbol{x} \in \mathbb{R}^n$, then $\psi$ is lower semicontinuous on $\mathbb{R}^n$ [31].

**Definition 1 (Subdifferentials and critical point)** *Let $\psi : \mathbb{R}^n \to \overline{\mathbb{R}}$ be a proper lower semicontinuous function.*

  $(i)$ *For each $\boldsymbol{x} \in \mathrm{dom}\,\psi$, the Fréchet subdifferential of $\psi$ at $\boldsymbol{x}$, written by $\hat{\partial}\psi(\boldsymbol{x})$, is the set of all vectors $\boldsymbol{u} \in \mathbb{R}^n$ which satisfy*

$$\liminf_{\substack{\boldsymbol{y} \to \boldsymbol{x} \\ \boldsymbol{y} \neq \boldsymbol{x}}} \frac{\psi(\boldsymbol{y}) - \psi(\boldsymbol{x}) - \langle \boldsymbol{u}, \boldsymbol{y} - \boldsymbol{x} \rangle}{\|\boldsymbol{y} - \boldsymbol{x}\|_2} \geqslant 0.$$

  *When $\boldsymbol{x} \notin \mathrm{dom}\,\psi$, we set $\hat{\partial}\psi(\boldsymbol{x}) = \varnothing$.*

  $(ii)$ *The limiting-subdifferential, or simply the subdifferential of $\psi$ at $\boldsymbol{x} \in \mathrm{dom}\,\psi$, written by $\partial\psi(\boldsymbol{x})$, is defined through the following closure process*

$$\partial\psi(\boldsymbol{x}) := \{\boldsymbol{u} \in \mathbb{R}^n | \exists \boldsymbol{x}^k \to \boldsymbol{x}, \ \psi(\boldsymbol{x}^k) \to \psi(\boldsymbol{x}) \ and \ \boldsymbol{u}^k \in \hat{\partial}\psi(\boldsymbol{x}^k) \to \boldsymbol{u} \ as \ k \to +\infty\}.$$

*We call an element in $\partial\psi(\boldsymbol{x})$ subgradient of $\psi$ at $\boldsymbol{x}$. We say that $\boldsymbol{x}$ is a critical point of $\psi$ if $\boldsymbol{0}_n \in \partial\psi(\boldsymbol{x})$.*

We also recall the following known results about subdifferential from Theorem 8.6, Exercise 8.8 (c) and Theorem 10.1 of [31], respectively.

**Fact 12** *For $\boldsymbol{x} \in \mathrm{dom}\, \psi$, $\hat{\partial}\psi(\boldsymbol{x}) \subset \partial\psi(\boldsymbol{x})$.*

**Fact 13** *Let $\psi_1 : \mathbb{R}^n \to \overline{\mathbb{R}}$ and $\psi_2 : \mathbb{R}^n \to \overline{\mathbb{R}}$ be two proper lower semicontinuous functions and $\boldsymbol{x} \in \mathbb{R}^n$. If $\psi_1$ is differentiable on a neighborhood of $\boldsymbol{x}$ and $\psi_2$ is finite at $\boldsymbol{x}$, then*

$$\partial(\psi_1 + \psi_2)(\boldsymbol{x}) = \nabla\psi_1(\boldsymbol{x}) + \partial\psi_2(\boldsymbol{x}).$$

**Fact 14 (Fermat's rule)** *If $\boldsymbol{x} \in \mathbb{R}^n$ is a local minimizer of $\psi$, then $\mathbf{0}_n \in \partial\psi(\boldsymbol{x})$.*

We shall use Theorem 2.9 of [4], which is recalled as Proposition 15, to prove the convergence of the PGA. For this purpose, we recall the notions of Kurdyka-Łojasiewicz (KL) property and KL function.

**Definition 2 (KL property)** *Let $\psi : \mathbb{R}^n \to \overline{\mathbb{R}}$ be a proper semicontinuous function. We say that $\psi$ satisfies the KL property at $\hat{\boldsymbol{x}} \in \mathrm{dom}\, \partial\psi$ if there exist $\eta \in (0, +\infty]$, a neighborhood $U$ of $\hat{\boldsymbol{x}}$ and a continuous concave function $\varphi : [0, \eta) \to [0, +\infty]$ such that*

(i) $\varphi(0) = 0$;

(ii) $\varphi$ is continuously differentiable on $(0, \eta)$ with $\varphi' > 0$;

(iii) $\varphi'(\psi(\boldsymbol{x}) - \psi(\hat{\boldsymbol{x}})) \cdot \mathrm{dist}(0, \partial\psi(\boldsymbol{x})) \geqslant 1$ for any $\boldsymbol{x} \in U \cap \{\boldsymbol{x} \in \mathbb{R}^n : \psi(\hat{\boldsymbol{x}}) < \psi(\boldsymbol{x}) < \psi(\hat{\boldsymbol{x}}) + \eta\}$.

**Definition 3 (KL function)** *We call a proper lower semicontinuous function $\psi : \mathbb{R}^n \to \overline{\mathbb{R}}$ KL function if $\psi$ satisfies the KL property at all points in $\mathrm{dom}\, \partial\psi$.*

**Proposition 15** *Let $\psi : \mathbb{R}^n \to \overline{\mathbb{R}}$ be a proper lower semicontinuous function. Consider a sequence $\{\boldsymbol{x}^{(k)}\}_{k \in \mathbb{N}} \subset \mathbb{R}^n$ satisfying the following conditions:*

(i) *There exists $a > 0$ such that*

$$\psi(\boldsymbol{x}^{(k+1)}) + a\|\boldsymbol{x}^{(k+1)} - \boldsymbol{x}^{(k)}\|_2^2 \leqslant \psi(\boldsymbol{x}^{(k)}), \ \text{for all } k \in \mathbb{N}.$$

(ii) *There exist $b > 0$ and $\boldsymbol{y}^{(k+1)} \in \partial\psi(\boldsymbol{x}^{(k+1)})$ such that*

$$\|\boldsymbol{y}^{(k+1)}\|_2 \leqslant b\|\boldsymbol{x}^{(k+1)} - \boldsymbol{x}^{(k)}\|_2, \ \text{for all } k \in \mathbb{N}.$$

(iii) *There exist a subsequence $\{\boldsymbol{x}^{(k_j)}\}_{j \in \mathbb{N}_+}$ and $\boldsymbol{x}^* \in \mathbb{R}^n$ such that*

$$\lim_{j \to \infty} \boldsymbol{x}^{(k_j)} = \boldsymbol{x}^* \ \text{ and } \ \lim_{j \to \infty} \psi(\boldsymbol{x}^{(k_j)}) = \psi(\boldsymbol{x}^*).$$

*If $\psi$ satisfies the KL property at $\boldsymbol{x}^*$, then*

$$\lim_{k \to \infty} \boldsymbol{x}^{(k)} = \boldsymbol{x}^* \text{ and } \mathbf{0}_n \in \partial\psi(\boldsymbol{x}^*).$$

We then focus on verifying that the sequence $\{\boldsymbol{v}^{(k)}\}_{k \in \mathbb{N}}$ generated by PGA satisfies all the conditions in Proposition 15. The satisfaction of item $(i)$ has been shown in Proposition 4. We next consider the satisfaction of item $(ii)$ in Proposition 15.

**Proposition 16** *Let $\{\boldsymbol{v}^{(k)}\}_{k \in \mathbb{N}}$ be generated by PGA. Then there exist $\boldsymbol{q}^{(k+1)} \in \partial F(\boldsymbol{v}^{(k+1)})$ and $b > 0$ such that*

$$\|\boldsymbol{q}^{(k+1)}\|_2 \leqslant b\|\boldsymbol{v}^{(k+1)} - \boldsymbol{v}^{(k)}\|_2, \ \text{for } k \in \mathbb{N}. \tag{A.16}$$

*Proof.* Let

$$\boldsymbol{q}^{(k+1)} := \frac{1}{\alpha}(\boldsymbol{v}^{(k)} - \boldsymbol{v}^{(k+1)}) + \nabla f(\boldsymbol{v}^{(k+1)}) - \nabla f(\boldsymbol{v}^{(k)}), \ \ k \in \mathbb{N},$$

and function $\varphi$ be defined by (A.12). We first prove that $\boldsymbol{q}^{(k+1)} \in \partial F(\boldsymbol{v}^{(k+1)})$. It follows from (3.9) and Fact 14 that $\boldsymbol{0}_N \in \partial\varphi(\boldsymbol{v}^{(k+1)})$, which together with Fact 13 yields that

$$\boldsymbol{v}^{(k)} - \boldsymbol{v}^{(k+1)} - \alpha\nabla f(\boldsymbol{v}^{(k)}) \in \partial\iota_\Omega(\boldsymbol{v}^{(k+1)}), \quad \text{for all } k \in \mathbb{N}.$$

Note that $\iota_\Omega = \alpha\iota_\Omega$. The above inclusion relation can be rewritten as

$$\frac{1}{\alpha}(\boldsymbol{v}^{(k)} - \boldsymbol{v}^{(k+1)}) - \nabla f(\boldsymbol{v}^{(k)}) \in \partial\iota_\Omega(\boldsymbol{v}^{(k+1)}), \quad \text{for all } k \in \mathbb{N}. \tag{A.17}$$

Now combining (A.17) and the fact $\partial F(\boldsymbol{v}^{(k+1)}) = \nabla f(\boldsymbol{v}^{(k+1)}) + \partial\iota_\Omega(\boldsymbol{v}^{(k+1)})$ yields that $\boldsymbol{q}^{(k+1)} \in \partial F(\boldsymbol{v}^{(k+1)})$, $k \in \mathbb{N}$.

We next prove that (A.16) holds. Since $\nabla f$ is $\boldsymbol{Q}_\epsilon$-Lipschitz continuous,

$$\|\boldsymbol{q}^{(k+1)}\|_2 \leqslant \frac{1}{\alpha}\|\boldsymbol{v}^{(k+1)} - \boldsymbol{v}^{(k)}\|_2 + \|\nabla f(\boldsymbol{v}^{(k+1)}) - \nabla f(\boldsymbol{v}^{(k)})\|_2 \leqslant b\|\boldsymbol{v}^{(k+1)} - \boldsymbol{v}^{(k)}\|_2,$$

where $b := \left(\frac{1}{\alpha} + \|\boldsymbol{Q}_\epsilon\|_2\right)$. This completes the proof. $\qquad\square$

We then consider the satisfaction of item $(iii)$ in Proposition 15. To this end, we need the following two lemmas.

**Lemma 17** *Let $\{\boldsymbol{v}^{(k)}\}_{k\in\mathbb{N}}$ be generated by PGA. If $\alpha \in \left(0, \frac{1}{\|\boldsymbol{Q}_\epsilon\|_2}\right)$, then $\{\boldsymbol{v}^{(k)}\}_{k\in\mathbb{N}}$ is bounded.*

**Proof.** We let $\gamma := \|\boldsymbol{I} - \alpha\boldsymbol{Q}_\epsilon\|_2$, and denote by $\lambda_{\max}(\boldsymbol{Q}_\epsilon)$, $\lambda_{\min}(\boldsymbol{Q}_\epsilon)$ the maximum and the minimum eigenvalues of $\boldsymbol{Q}_\epsilon$, respectively. Since $\boldsymbol{Q}_\epsilon$ is symmetric positive definite and $\alpha < \frac{1}{\|\boldsymbol{Q}_\epsilon\|_2} = \frac{1}{\lambda_{\max}(\boldsymbol{Q}_\epsilon)}$, we have $\gamma = 1 - \alpha \cdot \lambda_{\min}(\boldsymbol{Q}_\epsilon) \in (0, 1)$. From Proposition 3, we are easy to see that $\|\text{prox}_{\iota_\Omega}(\boldsymbol{v})\|_2 \leq \|\boldsymbol{v}\|_2$ for all $\boldsymbol{v} \in \mathbb{R}^N$, which together with (3.9) yields that

$$\|\boldsymbol{v}^{(k+1)}\|_2 \leqslant \|\boldsymbol{v}^{(k)} - \alpha\nabla f(\boldsymbol{v}^{(k)})\|_2 = \|(\boldsymbol{I} - \alpha\boldsymbol{Q}_\epsilon)\boldsymbol{v}^{(k)} + \alpha\boldsymbol{p}\|_2 \leqslant \gamma\|\boldsymbol{v}^{(k)}\|_2 + \alpha\|\boldsymbol{p}\|_2,$$

for all $k \in \mathbb{N}$. The above inequality implies that

$$\|\boldsymbol{v}^{(k+1)}\|_2 \leqslant \gamma^{k+1}\|\boldsymbol{v}^{(0)}\|_2 + \alpha\|\boldsymbol{p}\|_2 \sum_{j=0}^{k} \gamma^j = \gamma^{k+1}\|\boldsymbol{v}^{(0)}\|_2 + \alpha\|\boldsymbol{p}\|_2 \cdot \frac{1 - \gamma^{k+1}}{1 - \gamma},$$

for all $k \in \mathbb{N}$. Therefore, $\{\boldsymbol{v}^{(k)}\}_{k\in\mathbb{N}}$ is bounded, since $\gamma \in (0, 1)$. $\qquad\square$

**Lemma 18** *Let $\{\boldsymbol{v}^{(k)}\}_{k\in\mathbb{N}}$ be generated by PGA. If $\boldsymbol{v}^*$ is an accumulation point of $\{\boldsymbol{v}^{(k)}\}_{k\in\mathbb{N}}$, then $\boldsymbol{v}^* \in \Omega$.*

**Proof.** Since $\boldsymbol{v}^*$ is an accumulation point of $\{\boldsymbol{v}^{(k)}\}_{k\in\mathbb{N}}$, there exists a subsequence $\{\boldsymbol{v}^{(k_j)}\}_{j\in\mathbb{N}_+}$ of $\{\boldsymbol{v}^{(k)}\}_{k\in\mathbb{N}}$ such that $\lim_{j\to\infty} \boldsymbol{v}^{(k_j)} = \boldsymbol{v}^*$. Note that the set $\Omega$ is closed and $\boldsymbol{v}^{(k_j)} \in \Omega$ for all $j \in \mathbb{N}_+$. Hence $\boldsymbol{v}^* \in \Omega$, which completes the proof. $\qquad\square$

**Proposition 19** *Let $\{\boldsymbol{v}^{(k)}\}_{k\in\mathbb{N}}$ be generated by PGA and $F := f + \iota_\Omega$. If $\alpha \in \left(0, \frac{1}{\|\boldsymbol{Q}_\epsilon\|_2}\right)$, then there exist a subsequence $\{\boldsymbol{v}^{(k_j)}\}_{j\in\mathbb{N}_+}$ of $\{\boldsymbol{v}^{(k)}\}_{k\in\mathbb{N}}$ and $\boldsymbol{v}^* \in \Omega$ such that*

$$\lim_{j\to\infty} \boldsymbol{v}^{(k_j)} = \boldsymbol{v}^* \quad \text{and} \quad \lim_{j\to\infty} F(\boldsymbol{v}^{(k_j)}) = F(\boldsymbol{v}^*).$$

**Proof.** It follows from Lemma 17 that $\{\boldsymbol{v}^{(k)}\}_{k\in\mathbb{N}}$ is bounded. So there exists a subsequence $\{\boldsymbol{v}^{(k_j)}\}_{j\in\mathbb{N}_+}$ of $\{\boldsymbol{v}^{(k)}\}_{k\in\mathbb{N}}$ converges to some $\boldsymbol{v}^* \in \mathbb{R}^N$. It follows from Lemma 18 that $\boldsymbol{v}^* \in \Omega$. By the continuity of $f$ on $\mathbb{R}^N$, we have $\lim_{j\to\infty} f(\boldsymbol{v}^{(k_j)}) = f(\boldsymbol{v}^*)$. We also know that $\iota_\Omega(\boldsymbol{v}^*) = \iota_\Omega(\boldsymbol{v}^{(k)}) = 0$ for all $k \in \mathbb{N}$. Therefore, $\lim_{j\to\infty} F(\boldsymbol{v}^{(k_j)}) = F(\boldsymbol{v}^*)$, which completes the proof. $\quad\square$

To employ Proposition 15 for the convergence of PGA. We still need to show that $F$ satisfies the KL property at $\boldsymbol{v}^*$. To this end, we recall the notions of semi-algebraic sets and functions, and recall a known result in [4, 9] that establishes the relation between semi-algebraic property and KL property as Lemma 20.

**Definition 4 (Semi-algebraic sets and functions)** *A subset $\mathcal{S} \subset \mathbb{R}^n$ is called real semi-algebraic if it can be represented by*

$$\mathcal{S} = \bigcup_{j=1}^{s} \bigcap_{i=1}^{t} \left\{ \boldsymbol{x} \in \mathbb{R}^n \,|\, p_{ij}(\boldsymbol{x}) = 0, q_{ij}(\boldsymbol{x}) < 0 \right\}, \tag{A.18}$$

*where $p_{ij}$ and $q_{ij}$ are real polynomial functions for $i \in \mathbb{N}_t$, $j \in \mathbb{N}_s$, for some $s, t \in \mathbb{N}_+$. A function $\psi : \mathbb{R}^n \to \overline{\mathbb{R}}$ is called semi-algebraic if its graph $\{(\boldsymbol{x}, \psi(\boldsymbol{x})) : \boldsymbol{x} \in \operatorname{dom} \psi\}$ is a semi-algebraic subset of $\mathbb{R}^{n+1}$.*

**Lemma 20** *Let $\psi : \mathbb{R}^n \to \overline{\mathbb{R}}$ be a proper lower semicontinuous function. If $\psi$ is semi-algebraic, then it satisfies the KL property at any point of $\operatorname{dom} \partial \psi := \{\boldsymbol{u} \in \mathbb{R}^n \,|\, \partial \psi(\boldsymbol{u}) \neq \varnothing\}$.*

**Proposition 21** *Let $F := f + \iota_\Omega$, where function $\iota_\Omega$ and $f$ are defined by (3.6) and (3.7), respectively. Then $\operatorname{dom} \partial F = \Omega$ and $F$ is semi-algebraic.*

**Proof.** We first prove that $\operatorname{dom} \partial F = \Omega$. Note that $\operatorname{dom} F = \Omega$ is closed, which together with the definition of limiting-subdifferential implies that $\operatorname{dom} \partial F \subset \Omega$. For any $\boldsymbol{v} \in \Omega$, it is easy to verify that $\boldsymbol{0}_N \in \hat{\partial} \iota_\Omega(\boldsymbol{v})$. Then we see from Fact 12 that $\boldsymbol{0}_N \in \partial \iota_\Omega(\boldsymbol{v})$. Now by Fact 13, we have $\nabla f(\boldsymbol{v}) \in \partial F(\boldsymbol{v})$, which means that $\partial F(\boldsymbol{v}) \neq \varnothing$, that is, $\boldsymbol{v} \in \operatorname{dom} \partial F$. Hence $\Omega \subset \operatorname{dom} \partial F$. This proves $\operatorname{dom} \partial F = \Omega$.

We then prove that $F$ is semi-algebraic. From the definition of semi-algebraic function, we are easy to see that the sum of semi-algebraic functions is still semi-algebraic. It is obvious that function $f$ is semi-algebraic. To prove that $F$ is semi-algebraic, it suffices to show that $\iota_\Omega$ is semi-algebraic. The graph of $\iota_\Omega$ is given by

$$\operatorname{gra} \iota_\Omega = \left\{ \boldsymbol{x} \in \mathbb{R}^{N+1} \,\big|\, \boldsymbol{x}_{1:N} \geqslant \boldsymbol{0}_N, \|\boldsymbol{x}_{1:N}\|_0 \leqslant m \text{ and } x_{N+1} = 0 \right\}, \tag{A.19}$$

where $\boldsymbol{x}_{1:N} := (x_1, x_2, \ldots, x_N)^\top$. Note that there are $K := \binom{N}{N-m}$ combinations to choose an index set with $(N-m)$ elements out of the set $\{1, 2, \ldots, N\}$. We denote these index sets with size $(N-m)$ by $J_1, J_2, \ldots, J_K$, and let $\tilde{J}_i := J_i \cup \{N+1\}$, $i \in \mathbb{N}_K$. Then the graph of $\iota_\Omega$ in (A.19) can be represented by

$$\operatorname{gra} \iota_\Omega = \bigcup_{j=1}^{K} \left[ \left( \bigcap_{i \in \tilde{J}_j} \left\{ \boldsymbol{x} \in \mathbb{R}^{N+1} \,\big|\, x_i = 0 \right\} \right) \bigcap \left( \bigcap_{i \notin \tilde{J}_j} \left\{ \boldsymbol{x} \in \mathbb{R}^{N+1} \,\big|\, -x_i \leqslant 0 \right\} \right) \right],$$

which implies that $\iota_\Omega$ is a semi-algebraic function. This completes the proof. $\qquad \square$

We show in the following proposition that the objective function $F := f + \iota_\Omega$ satisfies the KL property at any accumulation point of sequence $\{\boldsymbol{v}^{(k)}\}_{k \in \mathbb{N}}$.

**Proposition 22** *Let $\{\boldsymbol{v}^{(k)}\}_{k \in \mathbb{N}}$ be generated by PGA. If $\boldsymbol{v}^*$ is an accumulation point of $\{\boldsymbol{v}^{(k)}\}_{k \in \mathbb{N}}$, then $F$ satisfies the KL property at $\boldsymbol{v}^*$.*

**Proof.** Since $\boldsymbol{v}^*$ is an accumulation point of $\{\boldsymbol{v}^{(k)}\}_{k \in \mathbb{N}}$, it follows from Lemma 18 that $\boldsymbol{v}^* \in \Omega$. By Proposition 21, we know that $F$ is semi-algebraic and $\boldsymbol{v}^* \in \operatorname{dom} \partial F$. Thus the desired result follows from Lemma 20 immediately. $\qquad \square$

We then show the continuity of the proximity operator $\operatorname{prox}_{\iota_\Omega}$ in the following proposition, which is also required to prove the convergence of PGA.

**Proposition 23** *Let $\{\boldsymbol{x}^{(k)}\}_{k \in \mathbb{N}} \subset \mathbb{R}^N$ be a sequence converges to some $\boldsymbol{x}^* \in \Omega$, and let $\boldsymbol{h}^{(k)} = \operatorname{prox}_{\iota_\Omega}(\boldsymbol{x}^{(k)})$ for $k \in \mathbb{N}$ and $\boldsymbol{h}^* = \operatorname{prox}_{\iota_\Omega}(\boldsymbol{x}^*)$ be given according to Proposition 3. Then $\lim_{k \to \infty} \boldsymbol{h}^{(k)} = \boldsymbol{h}^*$.*

**Proof.** Since $\boldsymbol{x}^* \in \Omega$, $m_{\boldsymbol{x}^*} \leqslant m$. We first consider the case $m_{\boldsymbol{x}^*} = 0$, that is, $x_j^* \leqslant 0$ for all $j \in \mathbb{N}_N$. Then $\boldsymbol{h}^* = \boldsymbol{0}_N$. For all $\varepsilon > 0$, we let $\delta_1 := \frac{\varepsilon}{\sqrt{N}}$. Then there exists $K_1 \in \mathbb{N}$ such that $x_j^{(k)} \leqslant \delta_1$ for

all $j \in \mathbb{N}_N$ and $k \geqslant K_1$. By the definition of $\boldsymbol{h}^{(k)}$, we know that $0 \leqslant h_j^{(k)} \leqslant \delta_1$ for all $j \in \mathbb{N}_N$ and $k \geqslant K_1$. Hence

$$\|\boldsymbol{h}^{(k)} - \boldsymbol{h}^*\|_2 = \|\boldsymbol{h}^{(k)}\|_2 \leqslant \sqrt{N}\delta_1 = \varepsilon, \ \text{ for all } k \geqslant K_1,$$

which implies $\lim_{k \to \infty} \boldsymbol{h}^{(k)} = \boldsymbol{h}^*$.

We then consider the case $0 < m_{\boldsymbol{x}^*} \leqslant m$. In this case, the set $\{j \in \mathbb{N}_N \mid x_j^* > 0\}$ is nonempty. For all $\varepsilon > 0$, we let $\delta_2 := \min\left\{\frac{1}{3}x_{\text{min-pos}}^*, \frac{\varepsilon}{\sqrt{N}}\right\}$, where

$$x_{\text{min-pos}}^* := \min_{j \in \mathbb{N}_N}\{x_j^* \mid x_j^* > 0\}.$$

There exists $K_2 \in \mathbb{N}$ such that for all $k \geqslant K_2$, $\|\boldsymbol{x}^{(k)} - \boldsymbol{x}^*\|_2 \leqslant \delta_2$, which indicates that

$$x_j^{(k)} \geqslant x_j^* - \delta_2 \geqslant \frac{2}{3}x_{\text{min-pos}}^* > 0, \ \text{ for } j \in J_{pos}^{\boldsymbol{x}^*} \tag{A.20}$$

and

$$x_j^{(k)} \leqslant x_j^* + \delta_2 \leqslant \delta_2 \leqslant \frac{1}{3}x_{\text{min-pos}}^*, \ \text{ for } j \in \mathbb{N}_N \backslash J_{pos}^{\boldsymbol{x}^*}. \tag{A.21}$$

By the fact $m_{\boldsymbol{x}^*} \leqslant m$ and the definitions of $\boldsymbol{h}^{(k)}$ and $\boldsymbol{h}^*$, we can conclude from (A.20) and (A.21) that for all $k \geqslant K_2$,

$$h_j^{(k)} = x_j^{(k)}, \ h_j^* = x_j^*, \ \text{ for } j \in J_{pos}^{\boldsymbol{x}^*}$$

and

$$0 \leqslant h_j^{(k)} \leqslant \delta_2, \ h_j^* = 0, \ \text{ for } j \in \mathbb{N}_N \backslash J_{pos}^{\boldsymbol{x}^*}.$$

Thus $\|\boldsymbol{h}^{(k)} - \boldsymbol{h}^*\|_2 \leqslant \sqrt{N}\delta_2 \leqslant \varepsilon$, which yields $\lim_{k \to \infty} \boldsymbol{h}^{(k)} = \boldsymbol{h}^*$. This completes the proof. $\square$

We are now in a position to utilize Proposition 15 and Theorem 2 to prove Theorem 5.

***Proof of Theorem 5.*** We first prove item $(i)$. According to Propositions 4, 16, 19 and 22, the convergence of $\{\boldsymbol{v}^{(k)}\}_{k \in \mathbb{N}}$ to a critical point $\boldsymbol{v}^* \in \Omega$ of $F := f + \iota_\Omega$ follows from Proposition 15 immediately. By item $(ii)$ in Theorem 2, to prove that $\boldsymbol{v}^*$ is a locally optimal solution of model (3.7) (or model (3.4)), it suffices to show that (3.8) holds. Since $\lim_{k \to \infty} \boldsymbol{v}^{(k)} = \boldsymbol{v}^*$, the Lipschitz continuity of $\nabla f$ yields that

$$\lim_{k \to \infty}\left(\boldsymbol{v}^{(k)} - \alpha\nabla f(\boldsymbol{v}^{(k)})\right) = \boldsymbol{v}^* - \alpha\nabla f(\boldsymbol{v}^*).$$

Now by letting $k \to \infty$ on both side of (3.9) and employing Proposition 23, we obtain (3.8), which proves the convergence of $\{\boldsymbol{v}^{(k)}\}_{k \in \mathbb{N}}$ to a locally optimal solution $\boldsymbol{v}^*$ of model (3.4).

We then prove the convergence rates of $\{\boldsymbol{v}^{(k)}\}_{k \in \mathbb{N}}$. Let $\Phi_k(\boldsymbol{v}) := \|\boldsymbol{v} - \boldsymbol{v}^{(k)} + \alpha\nabla f(\boldsymbol{v}^{(k)})\|_2^2$, $\boldsymbol{v} \in \mathbb{R}^N$. It is obvious that $\Phi_k$ is 2-strongly convex, since the Hessian matrix of function $\Phi_k - \|\cdot\|_2^2$ is positive semidefinite. To prove the convergence rate, we first show that there exists $K \in \mathbb{N}$ such that

$$\langle \nabla f(\boldsymbol{v}^*), \boldsymbol{v}^{(k)} - \boldsymbol{v}^* \rangle \geqslant 0 \ \text{ and } \ \langle \nabla\Phi_k(\boldsymbol{v}^{(k+1)}), \boldsymbol{v}^* - \boldsymbol{v}^{(k+1)} \rangle \geqslant 0, \tag{A.22}$$

for all $k \geqslant K$. From the proof of Theorem 2 in Appendix A.2, we see that (A.9) holds. It has been shown that $\lim_{k \to \infty} \boldsymbol{v}^{(k)} = \boldsymbol{v}^*$ and $\boldsymbol{v}^{(k)} \in \Omega$ for all $k \in \mathbb{N}$ (see item $(i)$ in Proposition 4). Then there exists $K_1 \in \mathbb{N}$ such that $\boldsymbol{v}^{(k)} \in B(\boldsymbol{v}^*; \delta) \cap \Omega$ for all $k \geqslant K_1$, which together with (A.9) implies that the first inequality in (A.22) holds for all $k \geqslant K_1$. According to the definition of $\Phi_k$ and (3.9), we see that

$$\boldsymbol{v}^{(k+1)} = \underset{\boldsymbol{v} \in \mathbb{R}^N}{\operatorname{argmin}} \frac{1}{2}\|\boldsymbol{v} - \boldsymbol{v}^{(k)} + \alpha\nabla f(\boldsymbol{v}^{(k)})\|_2^2 + \iota_\Omega(\boldsymbol{v}) = \underset{\boldsymbol{v} \in \Omega}{\operatorname{argmin}} \Phi_k(\boldsymbol{v}).$$

By a procedure similar to the first paragraph of the proof of Theorem 2, we can establish that

$$\boldsymbol{v}^{(k+1)} = \operatorname{prox}_{\iota_\Omega}\left(\boldsymbol{v}^{(k+1)} - \alpha\nabla\Phi_k(\boldsymbol{v}^{(k+1)})\right).$$

Note that $\Phi_k$ is also strongly convex. Using a proof analogous to that of the first inequality in (A.22), we can establish the existence of $K_2 \in \mathbb{N}$ such that the second inequality in (A.22) holds for all $k \geqslant K_2$. By setting $K = \max\{K_1, K_2\}$, we conclude that (A.22) holds for all $k \geqslant K$.

It follows from Lemma 11 that

$$\Phi_k(\boldsymbol{v}^*) \geqslant \Phi_k(\boldsymbol{v}^{(k+1)}) + \langle \nabla \Phi_k(\boldsymbol{v}^{(k+1)}), \boldsymbol{v}^* - \boldsymbol{v}^{(k+1)} \rangle + \|\boldsymbol{v}^* - \boldsymbol{v}^{(k+1)}\|_2^2, \tag{A.23}$$

which together with the second inequality in (A.22) implies that

$$\Phi_k(\boldsymbol{v}^*) \geqslant \Phi_k(\boldsymbol{v}^{(k+1)}) + \|\boldsymbol{v}^* - \boldsymbol{v}^{(k+1)}\|_2^2,$$

that is,

$$\|\boldsymbol{v}^* - \boldsymbol{v}^{(k)} + \alpha \nabla f(\boldsymbol{v}^{(k)})\|_2^2 \geqslant \|\boldsymbol{v}^{(k+1)} - \boldsymbol{v}^{(k)} + \alpha \nabla f(\boldsymbol{v}^{(k)})\|_2^2 + \|\boldsymbol{v}^* - \boldsymbol{v}^{(k+1)}\|_2^2, \tag{A.24}$$

for all $k \geqslant K$. For simplicity of notation, we define $\boldsymbol{z}^{(k)} := \boldsymbol{v}^{(k+1)} - \boldsymbol{v}^{(k)}$, $k \in \mathbb{N}$. Expanding (A.24) and dividing the resulting inequality by $2\alpha$ yields

$$\langle \nabla f(\boldsymbol{v}^{(k)}), \boldsymbol{z}^{(k)} \rangle + \frac{1}{2\alpha}\|\boldsymbol{z}^{(k)}\|_2^2 + \frac{1}{2\alpha}\|\boldsymbol{v}^* - \boldsymbol{v}^{(k+1)}\|_2^2 \leqslant \frac{1}{2\alpha}\|\boldsymbol{v}^* - \boldsymbol{v}^{(k)}\|_2^2 + \langle \nabla f(\boldsymbol{v}^{(k)}), \boldsymbol{v}^* - \boldsymbol{v}^{(k)} \rangle, \tag{A.25}$$

for all $k \geqslant K$. Recall that (A.14) in the proof of Proposition 4 holds. Since $\alpha \in \left(0, \frac{1}{\|\boldsymbol{Q}_\epsilon\|_2}\right)$, (A.14) implies that

$$f(\boldsymbol{v}^{(k+1)}) - f(\boldsymbol{v}^{(k)}) \leqslant \langle \nabla f(\boldsymbol{v}^{(k)}), \boldsymbol{z}^{(k)} \rangle + \frac{1}{2\alpha}\|\boldsymbol{z}^{(k)}\|_2^2, \quad \text{for all } k \in \mathbb{N}. \tag{A.26}$$

Combining (A.26) and (A.25), we obtain that

$$f(\boldsymbol{v}^{(k+1)}) - f(\boldsymbol{v}^{(k)}) + \frac{1}{2\alpha}\|\boldsymbol{v}^* - \boldsymbol{v}^{(k+1)}\|_2^2 \leqslant \frac{1}{2\alpha}\|\boldsymbol{v}^* - \boldsymbol{v}^{(k)}\|_2^2 + \langle \nabla f(\boldsymbol{v}^{(k)}), \boldsymbol{v}^* - \boldsymbol{v}^{(k)} \rangle, \tag{A.27}$$

for all $k \geqslant K$. It follows from the first order condition for convexity (Proposition B.3 of [7]) that

$$f(\boldsymbol{v}^*) \geqslant f(\boldsymbol{v}^{(k)}) + \langle \nabla f(\boldsymbol{v}^{(k)}), \boldsymbol{v}^* - \boldsymbol{v}^{(k)} \rangle,$$

which together with (A.27) yields

$$f(\boldsymbol{v}^{(k+1)}) + \frac{1}{2\alpha}\|\boldsymbol{v}^* - \boldsymbol{v}^{(k+1)}\|_2^2 \leqslant f(\boldsymbol{v}^*) + \frac{1}{2\alpha}\|\boldsymbol{v}^* - \boldsymbol{v}^{(k)}\|_2^2,$$

that is,

$$f(\boldsymbol{v}^{(k+1)}) - f(\boldsymbol{v}^*) \leqslant \frac{1}{2\alpha}\left(\|\boldsymbol{v}^{(k)} - \boldsymbol{v}^*\|_2^2 - \|\boldsymbol{v}^{(k+1)} - \boldsymbol{v}^*\|_2^2\right), \quad \text{for all } k \geqslant K.$$

We see from (A.11) that $\{f(\boldsymbol{v}^{(k)})\}_{k \in \mathbb{N}}$ is monotonically decreasing. Then for all $j \in \mathbb{N}_+$,

$$j\left(f(\boldsymbol{v}^{(K+j)}) - f(\boldsymbol{v}^*)\right) \leqslant \sum_{i=K}^{K+j-1}\left(f(\boldsymbol{v}^{(i+1)}) - f(\boldsymbol{v}^*)\right)$$

$$\leqslant \frac{1}{2\alpha}\sum_{i=K}^{K+j-1}\left(\|\boldsymbol{v}^{(i)} - \boldsymbol{v}^*\|_2^2 - \|\boldsymbol{v}^{(i+1)} - \boldsymbol{v}^*\|_2^2\right)$$

$$= \frac{1}{2\alpha}\left(\|\boldsymbol{v}^{(K)} - \boldsymbol{v}^*\|_2^2 - \|\boldsymbol{v}^{(K+j)} - \boldsymbol{v}^*\|_2^2\right).$$

Hence

$$f(\boldsymbol{v}^{(K+j)}) - f(\boldsymbol{v}^*) \leqslant \frac{1}{2\alpha j}\|\boldsymbol{v}^{(K)} - \boldsymbol{v}^*\|_2^2. \tag{A.28}$$

Note that $\|\boldsymbol{v}^{(K)} - \boldsymbol{v}^*\|_2^2$ is a constant. Let $k = K + j$ and $C := \frac{1}{\alpha}\|\boldsymbol{v}^{(K)} - \boldsymbol{v}^*\|_2^2$. Then (A.28) implies that

$$f(\boldsymbol{v}^{(k)}) - f(\boldsymbol{v}^*) \leqslant C \cdot \frac{1}{2j} \leqslant C \cdot \frac{1}{k}, \quad \text{for } j \geqslant K.$$

Thus

$$|f(\boldsymbol{v}^{(k)}) - f(\boldsymbol{v}^*)| = O\left(\frac{1}{k}\right). \tag{A.29}$$

Combining Lemma 11 and the first inequality in (A.22), we obtain that

$$f(\boldsymbol{v}^{(k)}) - f(\boldsymbol{v}^*) \geqslant \frac{\epsilon}{2}\|\boldsymbol{v}^{(k)} - \boldsymbol{v}^*\|_2^2, \quad \text{for all } k \geqslant K.$$

This together with (A.29) yields that $\|\boldsymbol{v}^{(k)} - \boldsymbol{v}^*\|_2 = O\left(\frac{1}{\sqrt{k}}\right)$.

We next prove item $(ii)$. The fact $\boldsymbol{v}^* \geqslant \boldsymbol{0}_N$ follows from (3.8) and Proposition 3 directly. Assume, to reach a contradiction, that $\boldsymbol{v}^* = \boldsymbol{0}_N$. Item $(i)$ in this theorem shows that $\boldsymbol{v}^*$ is a locally optimal solution of model (3.7). Then there exists $\delta > 0$ such that

$$f(\boldsymbol{v}) \geqslant f(\boldsymbol{v}^*) = f(\boldsymbol{0}_N) = 0, \quad \text{for all } \boldsymbol{v} \in \Omega \cap B(\boldsymbol{v}^*; \delta). \tag{A.30}$$

We recall from the assumption of this theorem that there exists $\tilde{\boldsymbol{w}} \in \Omega$ such that $\boldsymbol{p}^\top \tilde{\boldsymbol{w}} > 0$. Let $\tilde{\boldsymbol{w}}_\alpha := \alpha\tilde{\boldsymbol{w}}$, where $\alpha > 0$. Then $\tilde{\boldsymbol{w}}_\alpha \in \Omega$ and $\boldsymbol{p}^\top \tilde{\boldsymbol{w}}_\alpha > 0$. Note that when $\alpha$ tends to 0, the quadratic term $\frac{1}{2}\tilde{\boldsymbol{w}}_\alpha^\top \boldsymbol{Q}_\epsilon \tilde{\boldsymbol{w}}_\alpha$ is of higher order infinitesimal than the linear term $\boldsymbol{p}^\top \tilde{\boldsymbol{w}}_\alpha$. There exists some sufficient small $\alpha > 0$ such that $\tilde{\boldsymbol{w}}_\alpha \in B(\boldsymbol{v}^*; \delta)$ and

$$f(\tilde{\boldsymbol{w}}_\alpha) = \frac{1}{2}\tilde{\boldsymbol{w}}_\alpha^\top \boldsymbol{Q}_\epsilon \tilde{\boldsymbol{w}}_\alpha - \boldsymbol{p}^\top \tilde{\boldsymbol{w}}_\alpha < 0,$$

which contradicts (A.30). Therefore, $\boldsymbol{v}^* \neq \boldsymbol{0}_N$.

Lastly, we prove item $(iii)$. Since $\boldsymbol{v}^* \geqslant \boldsymbol{0}_N$ and $\boldsymbol{v}^* \neq \boldsymbol{0}_N$. There exit $\varepsilon_0 > 0$ and $K' \in \mathbb{N}$ such that $(\boldsymbol{v}^*)^\top \boldsymbol{1}_N > \varepsilon_0$ and $(\boldsymbol{v}^{(k)})^\top \boldsymbol{1}_N > \varepsilon_0$ for all $k \geqslant K'$, and hence $\boldsymbol{w}^{(k)} = \frac{\boldsymbol{v}^{(k)}}{(\boldsymbol{v}^{(k)})^\top \boldsymbol{1}_N}$ for all $k \geqslant K'$. Note that $\boldsymbol{v}^*$ and $\{\boldsymbol{v}^{(k)}\}_{k \in \mathbb{N}}$ are both bounded. There exist $C_1 > 0$ and $C_2 > 0$ such that

$$\frac{\|\boldsymbol{v}^{(k)}\|_2}{\left|(\boldsymbol{v}^{(k)})^\top \boldsymbol{1}_N\right| \cdot |(\boldsymbol{v}^*)^\top \boldsymbol{1}_N|} \leqslant C_1 \quad \text{and} \quad C_1\sqrt{N} + \left|\frac{1}{(\boldsymbol{v}^*)^\top \boldsymbol{1}_N}\right| \leqslant C_2.$$

Then for all $k \geqslant K'$,

$$\begin{aligned}
\left\|\boldsymbol{w}^{(k)} - \frac{\boldsymbol{v}^{(k)}}{(\boldsymbol{v}^*)^\top \boldsymbol{1}_N}\right\|_2 &\leqslant \left|\frac{1}{(\boldsymbol{v}^{(k)})^\top \boldsymbol{1}_N} - \frac{1}{(\boldsymbol{v}^*)^\top \boldsymbol{1}_N}\right| \cdot \|\boldsymbol{v}^{(k)}\|_2 \\
&= \frac{\|\boldsymbol{v}^{(k)}\|_2}{\left|(\boldsymbol{v}^{(k)})^\top \boldsymbol{1}_N\right| \cdot |(\boldsymbol{v}^*)^\top \boldsymbol{1}_N|} \cdot \left|(\boldsymbol{v}^{(k)} - \boldsymbol{v}^*)^\top \boldsymbol{1}_N\right| \\
&\leqslant C_1\sqrt{N}\|\boldsymbol{v}^{(k)} - \boldsymbol{v}^*\|_2,
\end{aligned}$$

and hence

$$\begin{aligned}
\|\boldsymbol{w}^{(k)} - \boldsymbol{w}^*\|_2 &= \left\|\boldsymbol{w}^{(k)} - \frac{\boldsymbol{v}^{(k)}}{(\boldsymbol{v}^*)^\top \boldsymbol{1}_N} + \frac{\boldsymbol{v}^{(k)}}{(\boldsymbol{v}^*)^\top \boldsymbol{1}_N} - \boldsymbol{w}^*\right\|_2 \\
&\leqslant C_1\sqrt{N}\|\boldsymbol{v}^{(k)} - \boldsymbol{v}^*\|_2 + \left|\frac{1}{(\boldsymbol{v}^*)^\top \boldsymbol{1}_N}\right|\|\boldsymbol{v}^{(k)} - \boldsymbol{v}^*\|_2 \\
&\leqslant C_2\|\boldsymbol{v}^{(k)} - \boldsymbol{v}^*\|_2.
\end{aligned}$$

This implies that $\|\boldsymbol{w}^{(k)} - \boldsymbol{w}^*\|_2 = O\left(\frac{1}{\sqrt{k}}\right)$. Similarly, we can prove that there exists $C_3 > 0$ such that

$$|S(\boldsymbol{w}^{(k)}) - S(\boldsymbol{w}^*)| \leqslant C_3\|\boldsymbol{w}^{(k)} - \boldsymbol{w}^*\|_2, \quad \text{for all } k \geqslant K',$$

which implies that $|S(\boldsymbol{w}^{(k)}) - S(\boldsymbol{w}^*)| = O\left(\frac{1}{\sqrt{k}}\right)$. This completes the proof. $\qquad\square$

## A.6 Proof of Theorem 6

To prove Theorem 6, we recall Proposition 11.4 of [6] as the following lemma.

**Lemma 24** *Let $\psi : \mathbb{R}^n \to \overline{\mathbb{R}}$ be be proper and convex. Then every local minimizer of $\psi$ is a global minimizer.*

**_Proof of Theorem 6._** We first prove item $(i)$. Let $\iota_{\hat\Omega}$ be defined by

$$\iota_{\hat\Omega}(\boldsymbol{v}) := \begin{cases} 0, & if\ \boldsymbol{v} \in \hat\Omega; \\ +\infty, & otherwise. \end{cases}$$

Then model (3.10) is equivalent to $\min\limits_{\boldsymbol{v}\in\mathbb{R}^N} \hat{F}(\boldsymbol{v})$, where $\hat{F} := f + \iota_{\hat\Omega}$. Of course, $\iota_{\hat\Omega}$ is proper. The convexity of $\hat\Omega$ implies that $\iota_{\hat\Omega}$ is convex (see Example 8.3 of [6]). Recall that $f$ is strictly convex, and hence $\hat{F}$ is proper and strictly convex. Since $\boldsymbol{v}^*$ is a locally optimal solution of model (3.7) and $\hat\Omega \subset \Omega$, there exists $\delta > 0$ such that

$$f(\boldsymbol{u}) \geqslant f(\boldsymbol{v}^*), \quad \text{for all } \boldsymbol{u} \in B(\boldsymbol{v}^*;\delta) \cap \hat\Omega. \tag{A.31}$$

The fact $\boldsymbol{v}^* \in \hat\Omega$ gives $\iota_{\hat\Omega}(\boldsymbol{v}^*) = 0$. Then (A.31) implies that $\hat{F}(\boldsymbol{u}) \geqslant \hat{F}(\boldsymbol{v}^*)$ for all $\boldsymbol{u} \in B(\boldsymbol{v}^*;\delta)$, that is, $\boldsymbol{v}^*$ is a local minimizer of $\hat{F}$. Now it follows from Lemma 24 that $\boldsymbol{v}^*$ is also a global minimizer of $\hat{F}$. The strict convexity of $\hat{F}$ implies the uniqueness of its global minimizer.

We next prove item $(ii)$. From item $(i)$ in this theorem and item $(ii)$ in Theorem 5, we see that $\boldsymbol{v}^* \in \hat\Omega$ is a globally optimal solution of model (3.10) and $\boldsymbol{v}^* \neq \boldsymbol{0}_N$. Then we are able to prove that $\boldsymbol{p}^\top \boldsymbol{v}^* = (\boldsymbol{v}^*)^\top \boldsymbol{Q}_\epsilon \boldsymbol{v}^* > 0$. We omit this proof here since it is very similar to the proof of Lemma 8. Now we have $\boldsymbol{v}^* = \frac{\boldsymbol{p}^\top \boldsymbol{v}^*}{(\boldsymbol{v}^*)^\top \boldsymbol{Q}_\epsilon \boldsymbol{v}^*} \boldsymbol{v}^*$. For any $\boldsymbol{v} \in \hat\Omega$ such that $\boldsymbol{p}^\top \boldsymbol{v} > 0$, we let $\boldsymbol{u} := \frac{\boldsymbol{p}^\top \boldsymbol{v}}{\boldsymbol{v}^\top \boldsymbol{Q}_\epsilon \boldsymbol{v}} \boldsymbol{v}$. Then $\boldsymbol{u} \in \hat\Omega$ and $\boldsymbol{p}^\top \boldsymbol{u} > 0$. Hence

$$-\frac{1}{2}\frac{(\boldsymbol{p}^\top \boldsymbol{v}^*)^2}{(\boldsymbol{v}^*)^\top \boldsymbol{Q}_\epsilon \boldsymbol{v}^*} = \frac{1}{2}(\boldsymbol{v}^*)^\top \boldsymbol{Q}_\epsilon \boldsymbol{v}^* - \boldsymbol{p}^\top \boldsymbol{v}^* \leqslant \frac{1}{2}\boldsymbol{u}^\top \boldsymbol{Q}_\epsilon \boldsymbol{u} - \boldsymbol{p}^\top \boldsymbol{u} = -\frac{1}{2}\frac{(\boldsymbol{p}^\top \boldsymbol{v})^2}{\boldsymbol{v}^\top \boldsymbol{Q}_\epsilon \boldsymbol{v}},$$

which implies that

$$\frac{\boldsymbol{p}^\top \boldsymbol{v}^*}{\sqrt{(\boldsymbol{v}^*)^\top \boldsymbol{Q}_\epsilon \boldsymbol{v}^*}} \geqslant \frac{\boldsymbol{p}^\top \boldsymbol{v}}{\sqrt{\boldsymbol{v}^\top \boldsymbol{Q}_\epsilon \boldsymbol{v}}}, \quad \text{for all } \boldsymbol{v} \in \hat\Omega.$$

Since $\boldsymbol{v}^* \in \hat\Omega$ and $\boldsymbol{v}^* \neq \boldsymbol{0}_N$, by the definition of $\hat\Omega_1$, we see that $\boldsymbol{w}^* \in \hat\Omega_1$. Note that $\hat\Omega_1 \subset \hat\Omega$. For all $\boldsymbol{w} \in \hat\Omega_1$,

$$\frac{\boldsymbol{p}^\top \boldsymbol{w}}{\sqrt{\boldsymbol{w}^\top \boldsymbol{Q}_\epsilon \boldsymbol{w}}} \leqslant \frac{\boldsymbol{p}^\top \boldsymbol{v}^*}{\sqrt{(\boldsymbol{v}^*)^\top \boldsymbol{Q}_\epsilon \boldsymbol{v}^*}} = \frac{\boldsymbol{p}^\top \boldsymbol{w}^*}{\sqrt{(\boldsymbol{w}^*)^\top \boldsymbol{Q}_\epsilon \boldsymbol{w}^*}},$$

which implies that $\boldsymbol{w}^*$ is a globally optimal solution of model $\max\limits_{\boldsymbol{w}\in\hat\Omega_1} S(\boldsymbol{w})$.

Lastly, we prove item $(iii)$. Note that $w_j^* > 0$ for all $j \in J_{\text{pos}}^{\boldsymbol{v}^*}$. Let $w_{\text{min-pos}}^* := \min\limits_{j \in J_{\text{pos}}^{\boldsymbol{v}^*}} \{w_j^*\}$, $\delta := \frac{1}{3}w_{\text{min-pos}}^*$, and let $\boldsymbol{w}$ be any vector in $B(\boldsymbol{w}^*;\delta) \cap \Omega_1$. Then $w_j \geqslant 2\delta > 0$ for $j \in J_{\text{pos}}^{\boldsymbol{v}^*}$, and $|w_j| \leqslant \delta$ for $j \in \mathbb{N}_N \backslash J_{\text{pos}}^{\boldsymbol{v}^*}$. Since $\boldsymbol{w} \in \Omega_1$ and $m_{\boldsymbol{v}^*} = m$, we conclude that $w_j = 0$ for $j \in \mathbb{N}_N \backslash J_{\text{pos}}^{\boldsymbol{v}^*}$, which implies that $\boldsymbol{w} \in \hat\Omega_1$. It has been shown that $\boldsymbol{w}^*$ is an optimal solution of model $\max\limits_{\boldsymbol{w}\in\hat\Omega_1} S(\boldsymbol{w})$. Therefore, $S(\boldsymbol{w}) \leqslant S(\boldsymbol{w}^*)$ for all $\boldsymbol{w} \in B(\boldsymbol{w}^*;\delta) \cap \Omega_1$, that is, $\boldsymbol{w}^*$ is a locally optimal solution of model (3.3). This completes the proof. $\qquad\square$

### A.7 Proof of Theorem 7

**_Proof._** According to Theorem 1 and item $(i)$ in Theorem 2, to prove the desired result, it suffices to show that

$$\boldsymbol{v}^* = \text{prox}_{\iota_\Omega}\left(\boldsymbol{v}^* - \frac{1}{\epsilon}\nabla f(\boldsymbol{v}^*)\right). \tag{A.32}$$

From the proof of Theorem 5, we know that (3.8) holds. According to the computation of $\text{prox}_{\iota_\Omega}$ in Proposition 3, to guarantee the validity of (3.8), we have $\nabla_i f(\boldsymbol{v}^*) = 0$ for all $i \in \text{supp}(\boldsymbol{v}^*)$. Otherwise, there exists some $i_0 \in \text{supp}(\boldsymbol{v}^*)$ such that $v_{i_0}^* \neq v_{i_0}^* - \alpha\nabla_{i_0} f(\boldsymbol{v}^*)$, which together with Proposition 3 implies that $\boldsymbol{v}^* \neq \text{prox}_{\iota_\Omega}(\boldsymbol{v}^* - \alpha\nabla f(\boldsymbol{v}^*))$, a contradiction to (3.8).

Suppose that $m_{\boldsymbol{v}^*} < m$. Then we have $\nabla_i f(\boldsymbol{v}^*) \geqslant 0$ for all $i \in \mathbb{N}_N \backslash \text{supp}(\boldsymbol{v}^*)$. Otherwise, there exists some $i_1 \in \mathbb{N}_N \backslash \text{supp}(\boldsymbol{v}^*)$ such that $v_{i_1}^* - \alpha\nabla_{i_1} f(\boldsymbol{v}^*) > 0$. Note that $m_{\boldsymbol{v}^*} < m$. The operation

of $\mathrm{prox}_{\iota_\Omega}$ will preserve the positive value $v_{i_1}^* - \alpha\nabla_{i_1} f(\boldsymbol{v}^*)$ instead of truncating it as 0, which violates (3.8). In this case, we now have $v_i^* - \frac{1}{\epsilon}\nabla_i f(\boldsymbol{v}^*) = v_i^*$ for $i \in \mathrm{supp}(\boldsymbol{v}^*)$ and $v_i^* - \frac{1}{\epsilon}\nabla_i f(\boldsymbol{v}^*) \leqslant 0$ for $i \in \mathbb{N}_N\backslash\mathrm{supp}(\boldsymbol{v}^*)$, which imply that (A.32) holds.

Suppose that item $(ii)$ holds. Let $\delta := \min\{v_i^*|i \in \mathrm{supp}(\boldsymbol{v}^*)\} > 0$. For $i \in \mathbb{N}_N\backslash\mathrm{supp}(\boldsymbol{v}^*)$, since $\frac{1}{\epsilon}\nabla_i f(\boldsymbol{v}^*) > -\delta$ and $v_i = 0$, we have $v_i^* - \frac{1}{\epsilon}\nabla_i f(\boldsymbol{v}^*) < \delta$. Note that $m_{\boldsymbol{v}^*} = m$. The operation of $\mathrm{prox}_{\iota_\Omega}$ makes $v_i^* - \frac{1}{\epsilon}\nabla_i f(\boldsymbol{v}^*) = v_i^*$ for $i \in \mathrm{supp}(\boldsymbol{v}^*)$ and $v_i^* - \frac{1}{\epsilon}\nabla_i f(\boldsymbol{v}^*) = 0$ for $i \in \mathbb{N}_N\backslash\mathrm{supp}(\boldsymbol{v}^*)$, that is, (A.32) holds. This completes the proof. $\qquad\square$

### A.8 Validation of PGA's Global Optimality Through Simulation Experiments

To test the validation of PGA's global optimality, we conduct a set of simulation experiments by considering model (3.7), where $\boldsymbol{Q}_\epsilon := \boldsymbol{Q}^\top\boldsymbol{Q} + \epsilon\boldsymbol{I}$. The iterative scheme of PGA for solving this model is given by (3.9) with $\alpha = \frac{0.99}{\|\boldsymbol{Q}\|_2}$.

In the simulation experiments, we set $\boldsymbol{\Sigma} \in \mathbb{R}^{10\times10}$ by $\Sigma_{ij} := 0.5^{|i-j|}$, and randomly generate a matrix $\boldsymbol{Q} \in \mathbb{R}^{50\times10}$ from the multivariate normal distribution, with mean vector $\boldsymbol{0}_{10}$ and covariance matrix $\boldsymbol{\Sigma}$. We set $\boldsymbol{p}$ as a random vector with components that are randomly generated numbers in the range $[-10, 10]$, and casually set $\epsilon = 0.001$ and the sparsity $m = 3$.

The direct exhaustive approach enumerates all possible support set configurations, totaling $C_{10}^3 = 120$ cases. In each case, we solve a 3-dimension quadratic programming problem. By comparing the optimal solutions corresponding to these 120 cases, we obtain the exact globally optimal solution of model (3.7). After that, we can evaluate the optimality of PGA's convergence.

For each experiment, we performed 500 iterations of PGA. To ensure the robustness of our findings, we used three different initializations: $\boldsymbol{0}_N$, $\boldsymbol{1}_N/N$ and $\boldsymbol{1}_N$. We repeated the experiments $10^4$ times for each initialization, with different $\boldsymbol{Q}$ and $\boldsymbol{p}$ in each run. We found that in over 7,200 of the $10^4$ trials, for any of the three initializations, both the normalized error of the iterative sequence $\|\boldsymbol{v}^{(k)} - \boldsymbol{v}^*\|_2/\|\boldsymbol{v}^*\|_2$ and the normalized error of the function value $|f(\boldsymbol{v}^{(k)}) - f(\boldsymbol{v}^*)|/|f(\boldsymbol{v}^*)|$ were smaller than $10^{-10}$. Here $\boldsymbol{v}^*$ denotes the globally optimal solution, and $\boldsymbol{v}^{(k)}$ represents the iterative sequence at the $k$-th iteration of PGA. We show the plots of $\|\boldsymbol{v}^{(k)} - \boldsymbol{v}^*\|_2/\|\boldsymbol{v}^*\|_2$ and $|f(\boldsymbol{v}^{(k)}) - f(\boldsymbol{v}^*)|/|f(\boldsymbol{v}^*)|$ in the following Figure 1, and show in Table 4 these two normalized errors obtained at 500 iterations of PGA in ten simulation experiments.

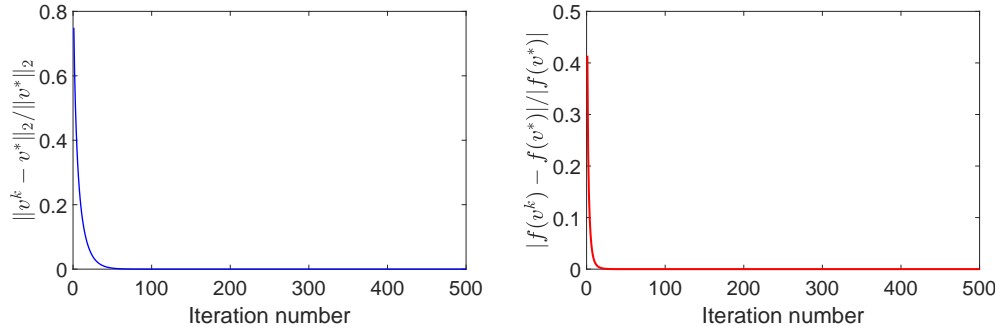

Figure 1: Simulation results of PGA for model (3.7). Left: normalized error of the iterative sequence versus number of iterations. Right: normalized error of function value versus number of iterations.

Table 4: The normalized errors obtained at 500 iterations of PGA in 10 simulation experiments.

| $k$ | 1 | 2 | 3 | 4 | 5 | 6 | 7 | 8 | 9 | 10 |
|---|---|---|---|---|---|---|---|---|---|---|
| $\frac{\|\boldsymbol{v}^{(k)}-\boldsymbol{v}^*\|_2}{\|\boldsymbol{v}^*\|_2}$ | 2.45E-16 | 1.62E-16 | 3.03E-16 | 6.23E-01 | 1.14E-16 | 7.68E-16 | 7.41E-16 | 9.55E-17 | 1.50E-16 | 2.52E-16 |
| $\frac{\|f(\boldsymbol{v}^{(k)})-f(\boldsymbol{v}^*)\|_2}{\|f(\boldsymbol{v}^*)\|_2}$ | 0.00 | 1.48E-16 | 1.68E-16 | 1.17E-01 | 0.00 | 1.40E-16 | 4.73E-16 | 1.36E-16 | 0.00 | 0.00 |

From the simulation experiments, we conclude that the proposed PGA has a high probability (over 72%) of directly converging to a globally optimal solution of model (3.7). This finding is consistent with the sufficient conditions for global optimality in Theorem 7.

## A.9 Solving Algorithm: mSSRM-PGA

---

**Algorithm A1** mSSRM-PGA

---

**Input:** Given the sample asset return matrix $\boldsymbol{R} \in \mathbb{R}^{T \times N}$ and the positive parameter $\epsilon$.

**Preparation:** Let $\boldsymbol{p} = \frac{1}{T}\boldsymbol{R}^{\top}\boldsymbol{1}_T$, $\boldsymbol{Q} = \frac{1}{\sqrt{T-1}}\left(\boldsymbol{R} - \frac{1}{T}\boldsymbol{1}_{T \times T}\boldsymbol{R}\right)$ and $\boldsymbol{Q}_\epsilon = \boldsymbol{Q}^{\top}\boldsymbol{Q} + \epsilon\boldsymbol{I}$. Compute the largest eigenvalue $\lambda_1$ of $\boldsymbol{Q}_\epsilon$, and set $\alpha = \frac{0.999}{\lambda_1}$.

**Initialization:** Set $\boldsymbol{v}^{(0)} = \boldsymbol{p}$, $tol = 10^{-5}$, $MaxIter = 10^4$ and $k = 0$.

**repeat**

1. $\boldsymbol{v}^{(k+1)} = \mathrm{prox}_{\iota_\Omega}\left(\boldsymbol{v}^{(k)} - \alpha\left(\boldsymbol{Q}_\epsilon\boldsymbol{v}^{(k)} - \boldsymbol{p}\right)\right)$

2. $k = k + 1$

**until** $\frac{\|\boldsymbol{v}^{(k)} - \boldsymbol{v}^{(k-1)}\|_2}{\|\boldsymbol{v}^{(k-1)}\|_2} \leqslant tol$ or $k > MaxIter$.

**if** $\boldsymbol{v}^{(k)} \neq \boldsymbol{0}_N$

3. $\boldsymbol{w}^* = \frac{\boldsymbol{v}^{(k)}}{(\boldsymbol{v}^{(k)})^{\top}\boldsymbol{1}_N}$

**else**

4. $\boldsymbol{w}^* = \boldsymbol{0}_N$

**Output**: The portfolio $\boldsymbol{w}^*$.

---

## A.10 Additional Experimental Results

The 1/N strategy rebalances to the equally-weighted portfolio on each trading time. S1, S2 and S3 are different versions of SSPO-$\ell_0$ in (2.5), among which S1 is deterministic but S2 and S3 are randomized. The hyperparameters of these competitors are set according to the original papers.

FF25 contains 25 portfolios developed by BE/ME (book equity to market equity) and investment in the US market. FF25EU contains 25 portfolios developed by ME and prior return in the European market. FF32 contains 32 portfolios developed by BE/ME and investment in the US market. FF49 contains 49 industry portfolios in the US market. FF100 contains 100 portfolios developed by ME and BE/ME, while FF100MEINV contains 100 portfolios developed by ME and investment, both in the US market. The information of these data sets are given in Table 5.

Table 5: Information of 6 real-world monthly benchmark data sets.

| Data Set | Region | Time | Months | Assets |
|---|---|---|---|---|
| FF25 | US | $Jul/1971 \sim May/2023$ | 623 | 25 |
| FF25EU | EU | $Nov/1990 \sim May/2023$ | 391 | 25 |
| FF32 | US | $Jul/1971 \sim May/2023$ | 623 | 32 |
| FF49 | US | $Jul/1971 \sim May/2023$ | 623 | 49 |
| FF100 | US | $Jul/1971 \sim May/2023$ | 623 | 100 |
| FF100MEINV | US | $Jul/1971 \sim May/2023$ | 623 | 100 |

There is a relaxation approach based on the semi-definite programming (SDP Relaxation, [20]) that intends to address nearly the same mSSRM model (3.3) of this paper, except for relaxing the cardinality constraint and the long-only constraint. Therefore, this method fails to control cardinality exactly and a simplex projection [16] should be implemented to ensure feasibility. Its experimental results are also provided in Table 6, which are not so good as those of mSSRM-PGA.

Table 6: Final cumulative wealths (CW) and Sharpe Ratios (SR) of SDP Relaxation and mSSRM-PGA on 6 data sets ($T = 60$).

| Data Set | FF25 | | FF25EU | | FRENCH32 | | FF49 | | FF100 | | FF100MEINV | |
|---|---|---|---|---|---|---|---|---|---|---|---|---|
| Strategy | CW | SR | CW | SR | CW | SR | CW | SR | CW | SR | CW | SR |
| SDP Relaxation | 323.76 | 0.2340 | 14.25 | 0.1674 | 290.24 | 0.2224 | 280.46 | 0.2151 | 0.51 | 0.0218 | 194.09 | 0.1528 |
| **mSSRM-PGA (m=10)** | **615.34** | **0.2481** | **126.02** | **0.2712** | **991.89** | **0.2612** | **285.02** | **0.2151** | **527.09** | **0.2290** | **375.75** | **0.2217** |
| **mSSRM-PGA (m=15)** | **614.71** | **0.2481** | **125.19** | **0.2708** | **996.32** | **0.2615** | **262.54** | **0.2135** | **522.28** | **0.2289** | **383.44** | **0.2232** |
| **mSSRM-PGA (m=20)** | **614.70** | **0.2481** | **125.19** | **0.2708** | **996.23** | **0.2615** | **262.06** | **0.2134** | **515.50** | **0.2285** | **384.65** | **0.2234** |

As for practical issues, Table 7 shows the running times for different methods with $T = 60$, where mSSRM-PGA achieves competitive computational efficiency. Figure 2 shows the final cumulative wealths of different methods as the transaction cost rate $\nu$ varies from 0 to 0.5% with $T = 60$, which indicates that mSSRM-PGA can withstand considerable levels of transaction cost rates.

Table 7: The average running times (in seconds) of different portfolio optimization models for one period on 6 data sets.

| Data Set | SPOLC | SSPO | S1 | S2 | S3 | SSMP | MAXER | IPSRM-D | PLCT | SDP Relaxation | mSSRM-PGA |
|---|---|---|---|---|---|---|---|---|---|---|---|
| FF25 | 0.0263 | 0.0234 | 5.72E-05 | 5.63E-05 | 8.46E-05 | 0.0122 | 0.0525 | 0.0009 | 0.0020 | 1.0115 | 0.0052 |
| FF25EU | 0.0222 | 0.0239 | 1.70E-05 | 2.89E-05 | 2.59E-05 | 0.0316 | 0.0588 | 0.0009 | 0.0015 | 0.8178 | 0.0059 |
| FRENCH32 | 0.0239 | 0.0250 | 2.93E-05 | 2.81E-05 | 3.08E-05 | 0.0075 | 0.0392 | 0.0012 | 0.0021 | 1.2862 | 0.0075 |
| FF49 | 0.0252 | 0.0458 | 2.94E-05 | 3.51E-05 | 2.82E-05 | 0.0083 | 0.0270 | 0.0029 | 0.0034 | 12.3780 | 0.0114 |
| FF100 | 0.0306 | 0.0854 | 5.38E-05 | 4.48E-05 | 4.27E-05 | 0.0451 | 0.0458 | 0.0132 | 0.0052 | 24.5852 | 0.0713 |
| FF100MEINV | 0.0296 | 0.0864 | 5.08E-05 | 4.81E-05 | 4.53E-05 | 0.0145 | 0.0152 | 0.0144 | 0.0059 | 23.9911 | 0.0696 |

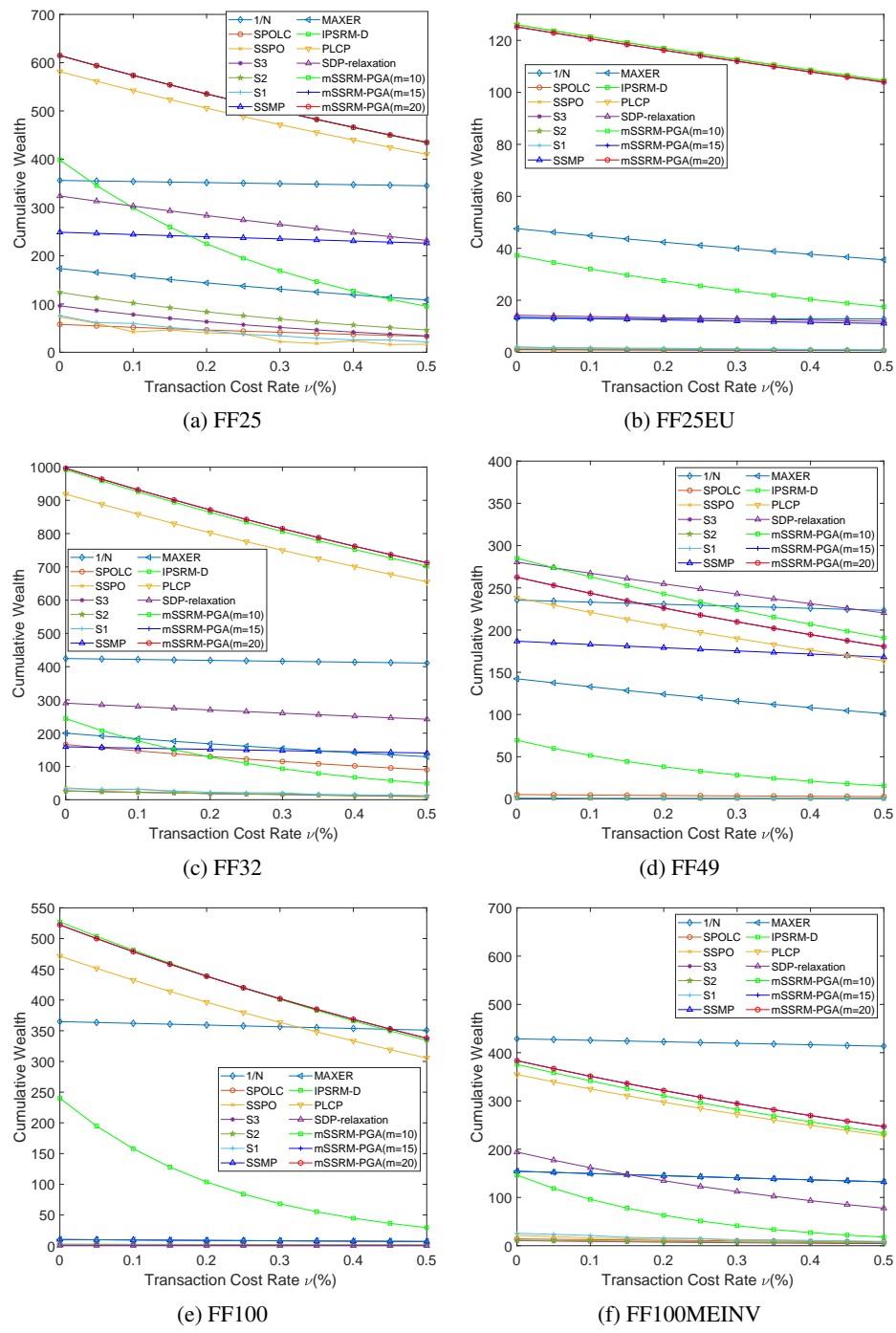

(a) FF25

(b) FF25EU

(c) FF32

(d) FF49

(e) FF100

(f) FF100MEINV

Figure 2: Final cumulative wealths of portfolio optimization methods w.r.t. transaction cost rate $\nu$ on 6 benchmark data sets.

