# OpenReview forum: "A Globally Optimal Portfolio for m-Sparse Sharpe Ratio Maximization"
_NeurIPS.cc/2024/Conference — NeurIPS 2024 poster_

### Official Review · Reviewer_pxYT · 2024-07-11

**Soundness:** 3
**Presentation:** 3
**Contribution:** 3
**Rating:** 5
**Confidence:** 4

**Summary:**

This study addresses the problem of Sharpe ratio maximization under a cardinality constraint, referred to in the paper as an m-sparse constraint. Adding a cardinality constraint typically makes optimization problems NP-hard. Existing studies usually approach this problem using heuristic methods or relaxations. The former can be time-consuming and do not guarantee optimality, while the latter struggles to control cardinality accurately. This research proposes transforming the m-sparse fractional optimization problem into an equivalent m-sparse quadratic programming problem, which ensures convergence to a locally optimal solution and allows for more precise control of cardinality. However, as will be discussed in the question section, the study essentially transforms the problem into a mean-variance optimization form. This raises the question of why not simply perform m-sparse mean-variance optimization. Since Sharpe ratio maximization is just one of many Pareto optimal points in mean-variance optimization, addressing the more general problem could be more advantageous. Also, as this paper deals with practical issue (i.e., cardinality constraint), the authors should discuss more about practical implementation of the proposed method (e.g., computational time, adding more practical constraints).

**Strengths:**

- The proposed method appears to be theoretically sound and effectively controls cardinality.
- The approach has been tested on various datasets, and the experimental results indicate strong performance.
- The writing structure is well-organized and the content is presented in a clear and readable manner.

**Weaknesses:**

- Ultimately, this research focuses on a portfolio optimization problem with a cardinality constraint driven by practical needs, making practical implementation the most critical aspect. However, there is a lack of discussion or analysis on this aspect.
- Practical implementation would require consideration of various additional constraints (e.g., lower or upper bound of portfolio weights, turnover constraint). It would be beneficial to demonstrate whether the proposed model can accommodate a range of additional constraints.
- There is a need for a comparative analysis of computation time to evaluate the efficiency of the proposed method.
- The following paper addresses the exact same problem of cardinality constrained Sharpe ratio maximization. While their approach, based on relaxation, has the advantage of transforming the problem into a convex optimization issue, it fails to control cardinality exactly.

[1] Kim, M. J., Lee, Y., Kim, J. H., & Kim, W. C. (2016). Sparse tangent portfolio selection via semi-definite relaxation. Operations Research Letters, 44(4), 540-543.

**Questions:**

The authors emphasize several times in bold that directly maximizing the Sharpe ratio is a key point of this study. However, as shown in equations (3.4) and (3.7), the problem is eventually transformed into a mean-variance optimization form, with Theorem 1 demonstrating how this can be converted back to a solution for the Sharpe ratio maximization problem. This raises the question of why the study is framed as a Sharpe ratio maximization problem. Why not simply present it as m-sparse mean-variance optimization, which is more general? After all, Sharpe ratio maximization is just one of many Pareto optimal points in mean-variance optimization. Is this framing due to overlapping aspects with previous research? Further clarification on this point would be helpful.

**Limitations:**

The authors mention the inability to directly apply fractional optimization as a limitation. However, I believe that more emphasis should be placed on practical implementation. Since this study defines and solves a problem based on practical needs, more attention should be given to practical implementation rather than purely mathematical formulation. Ensuring that the proposed method can be effectively implemented in real-world scenarios is crucial for its overall utility.

---

> ### Author Rebuttal · Authors · 2024-08-05
>
> **Answer for Weakness 1:**
>
> In modern portfolio management, it is widely-recognized that the number of selected assets should be restricted to a manageable size, in order to keep simplicity and save time and financial costs. Managerial strategies provide an approach to achieve this objective. However, the managerial approaches still require intensive administration and abundant experience in management and finance. Hence researchers turn to sparsity models for solutions via the computational approaches.
>
> **Answer for Weakness 2:**
>
> Lower or upper bound of portfolio weights, as well as the turnover constraint, can also be deployed in our method, as long as they are convex and have closed-form proximal mapping. One can replace the simplex constraint by the afore-mentioned constraints. The theory and the algorithm throughout the paper still hold.
>
> **Answer for Weakness 3:**
>
> We have conducted a rigorous theoretical analysis to conclusively demonstrate that the convergence rate of PGA in terms of the function value is $f(v^k)-f(v^*)=O(1/k)$. Since the proof is relatively lengthy, we did not include it here. In the upcoming revised version of the manuscript, we will incorporate the comprehensive proof to substantiate this finding. This convergence rate result implies that the computational complexity of the proposed PGA is $O(TN\varepsilon^{-1})$, where $T$ and $N$ denote the window size and the number of assets, respectively, $\varepsilon$ is the convergence tolerance in the objective function value.
>
> Response Table 2 in the attached PDF shows that the running time of mSSRM-PGA is competitive to those of the competitors. Hence mSSRM-PGA has high computational efficiency besides good investing performance.
>
> **Answer for Weakness 4:**
>
> This method fails to control cardinality exactly and actually solves another simplified optimization model. We add this method as comparison and the results in Response Table 3 show that it is not so good as ours.
>
> **Answer for Questions:**
>
> Please note that (3.4) is an equivalent optimization model to (3.3), but other general m-sparse mean-variance optimization models may not be. Solving a general m-sparse mean-variance optimization model may lead to a solution far from the locally optimal ones of (3.3). Therefore, we have carefully deduced (3.4) from (3.3) while preserving equivalence, which is a novel contribution.
>
> Additionally, we have now provided two important theoretical results beyond the original manuscript. First, we established more intuitive sufficient conditions for the proposed PGA algorithm to converge to a global optimal solution. Second, we proved that the convergence rate of PGA in terms of the function value is $f(v^k)-f(v^*)=O(1/k)$. For more details of these two results, please refer to Point 1 and Point 2 in the **'Author Rebuttal for Global Response'**. We believe that these new results significantly improve the completeness and contribution of our paper.
>
> We would like to add that the non-convex fractional model with the non-convex $\ell_0$-constraint presents a highly complex doubly non-convex optimization problem. Despite this complexity, our new result successfully provides theoretically verifiable sufficient conditions for convergence to the global optimum, which is a significant achievement. These rigorous theoretical results have meaningful implications for sparse multi-objective optimization in machine learning.

---

> > ### Comment · Reviewer_pxYT · 2024-08-11
> >
> > Thank you for the comments. But I still do not think directly solving 'cardinality constrained Sharpe ratio maximization problem' instead of 'cardinality constrained mean-variance optimization problems' is "novel". I will maintain my evaluation as it is.

---

### Official Review · Reviewer_4h7E · 2024-07-12

**Soundness:** 2
**Presentation:** 2
**Contribution:** 2
**Rating:** 5
**Confidence:** 3

**Summary:**

In summary, this paper studies Sharpe ratio optimization in portfolio management and contributes the achievement of sparse distribution iterates converging to a local optimum by converting the fractional optimization problem into a quadratic programming.

**Strengths:**

Originality:
The task of optimizing SR with sparse distributions is somewhat new.
The work incorporates quadratic programming and its well-known techniques into SR optimization in a novel way.

Quality:
The submission is technically sound.
The existing claims are generally supported by theory and experiments.
The methods are appropriate.

Clarity:
The writing of the submission is mostly clearly, well organized and adequately informs the reader.

Significance:
The work advances the state of the art in a demonstrable way with its theory.
Other researchers and practitioners are likely to use the ideas, possibly build on them, considering the experimental results.
With a somewhat unique theoretical approach, it provides unique conclusions about existing optimization targets in the form of sparse optimization.

**Weaknesses:**

Originality:
It is not clear how this work differs from previous contributions beyond the fact that they did not study this setting. Due to this, it is also possible that the related work is not adequately cited. More explanations are needed in that regard.

Quality:
This is more of a work in progress. The authors are at times not careful and possibly honest about evaluating their work. Their achievement is only to a local optimum and the feasibility of the scenario for global optimality is questionable. They also do not provide convergence rates, which is important in performance analysis.

Significance:
The importance of the results is a bit questionable due to the lack of specific comparisons with the literature. Hence, it is also not demonstrated that the submission addresses a difficult task in a better way than the previous works.

**Questions:**

Major Questions:
- Why are self-financing and long-only constraints needed?
- What does Section 2.1 accomplish? It is too superficial with no heed towards the roles of the introduced models and parameters in their respective optimization scenarios, akin to a laundry list of past works. How are they related to SR maximization? It appears each method aims for something different.
- What does "guarantee suboptimality" mean?
- Considering the definition of $Q_\epsilon$, when is (i) possible?
- What is the convergence rate?

Minor Questions:
- Page 1 Line 33: How is "market crashes" from [25] a strategy?

Suggestions:
- Please correct the reference numbering.
- Page 6 Line 240: Grammar error.

**Limitations:**

The authors addressed the limitations. For improvement, they are suggested to update their limitations with lack of convergence rate and global optimality.

---

> ### Author Rebuttal · Authors · 2024-08-05
>
> **Answer for Weakness 1 (Originality):**
>
> The crucial contribution of our work is maximizing SR under two constraint simultaneously: the m-sparse (cardinality) constraint and the simplex constraint. While there are indeed a bunch of works elaborating SR maximization under various constraints, few of them consider the former two constraints, especially the m-sparse constraint. This setting makes sense because the m-sparse constraint can accurately control the size of selected assets, and the simplex constraint ensures feasibility of the portfolio in practical issues.
>
> To further improve the completeness and contribution of our work, we have added three main components: 1. sufficient conditions for PGA's convergence to a global optimum; 2. analysis of the convergence rate of PGA; 3. validation of PGA's global optimality through simulation experiments. Please refer to **'Author Rebuttal for Global Response'** for more details.
>
> **Answer for Weakness 2 (Quality):**
>
> Our work is already a complete one that elaborates the best theoretical results on the addressed problem. We have already provided the loosest conditions under which local and even global optimality is guaranteed. Based on the existing results in the original manuscript, we have now proven more intuitive sufficient conditions for the convergence of PGA to a globally optimal solution of model (3.7). For a detailed proof, please refer to **1. Sufficient conditions for PGA's convergence to a global optimum** in the **'Author Rebuttal for Global Response'**. We also conducted a set of simulation experiments to show that the proposed PGA has a high probability (over 72%) of directly converging to a globally optimal solution. For more details of the simulation experiments, please refer to **3. Validation of PGA's global optimality through simulation experiments** in the **'Author Rebuttal for Global Response'**.
>
> Besides, we have conducted a rigorous theoretical analysis to conclusively demonstrate that the convergence rate of PGA in terms of the function value is $f(v^k)-f(v^*)=O(1/k)$. Since this proof is relatively lengthy, we did not include it here. In the upcoming revised version of the manuscript, we will incorporate the comprehensive proof to substantiate this finding.
>
> We would like to add that the non-convex fractional model with the non-convex $\ell_0$-constraint presents a highly complex doubly non-convex optimization problem. Despite this complexity, our new result successfully provides theoretically verifiable sufficient conditions for convergence to the global optimum, which is a significant achievement. These rigorous theoretical results have meaningful implications for sparse multi-objective optimization in machine learning.
>
> **Answer for Weakness 3 (Significance):**
>
> This is because few approaches are proposed to solve the proposed problem (3.3). We have to compare our method with other ones that solve simpler models like (2.6) and (2.8). For another example, [Kim et al., 2016] develops a semidefinite programming (SDP) relaxation of (3.3). As suggested by Reviewer pxYT, this method fails to control cardinality exactly and actually solves another simplified optimization model. We add this method as comparison and the results in Response Table 3 (see the attached PDF file) show that it is not so good as ours.
>
>
> **Answer for Major Questions:**
>
> (1) The self-financing constraint restricts the position that an investor can use to ensure feasibility. For example, suppose there are two assets A and B. An investor can eligibly invest 40\% and 60\% of the whole position on A and B, respectively. The self-financing constraint is (40\% + 60\%)=1 in this case. But if without a self-financing constraint, the investor may invest 50\% and 70\% on A and B, respectively. Then the whole position becomes (50\%+70\%)>1, which is infeasible. The long-only constraint means that an investor can only buy assets instead of short-selling, which is a conventional setting in portfolio optimization.
>
> (2) These are ordinary portfolio optimization models. Some of them are also competitors in the experiments. Since SR maximization originates from ordinary portfolio optimization, we provide a subsection for such background knowledge to the readers.
>
> (3) It means "guarantee the convergence to a locally optimal solution".
>
> (4) Thank you for mentioning the validity of the conditions in item $(i)$ of Theorem 2. Based on this question, we have proven more intuitive sufficient conditions for the convergence of PGA to a globally optimal solution of model (3.7). For a detailed proof, please refer to **1. Sufficient conditions for PGA's convergence to a global optimum** in the **'Author Rebuttal for Global Response'**.
>
> Specifically, if one of the following two conditions holds for the limit point $v^*$ of the sequence $\\{v^k\\}$ generated by PGA:
>
> (i) $\\|v^*\\|_0<m$,
>
> (ii) $\\|v^*\\|_0=m$ and $\nabla_i f(v^*)>-\epsilon\cdot\min\\{v_i^*|i\in{\rm supp}(v^*)\\}$ for all $i\in\mathbb{N}\backslash{\rm supp}(v^*)$,
>
> then the conditions in item $(i)$ of Theorem 2 holds, that is, $v^*$ is a globally optimal solution of of model (3.7).
>
> (5) We have conducted a rigorous theoretical analysis to conclusively demonstrate that the convergence rate of PGA in terms of the function value is $f(v^k)-f(v^*)=O(1/k)$. Since this proof is relatively lengthy, we did not include it here. In the upcoming revised version of the manuscript, we will incorporate the comprehensive proof to substantiate this finding.
>
>
> **Answer for Minor Questions:**
>
> It is a shorting (selling) based strategy that exploits market crash events.
>
>
> **Answer for Suggestions:**
>
> We appreciate the reviewer pointing out the issues with citation numbers and grammatical errors in the manuscript. We will thoroughly review the entire manuscript and make the necessary corrections.

---

> > ### Comment · Reviewer_4h7E · 2024-08-09
> >
> > Thank you for your answers. I am raising my score in consideration of the renewed global optimality claims.

---

### Official Review · Reviewer_EtEi · 2024-07-14

**Soundness:** 3
**Presentation:** 3
**Contribution:** 2
**Rating:** 5
**Confidence:** 3

**Summary:**

This paper studies the optimization of an m-sparse portfolio, which has an additional sparsity constraint on the portfolio compared to traditional portfolio optimization.
Instead of the mean-variance approach, this work proposed to directly optimize the fractional objective which is the Sharpe ratio. The paper shows that the m-sparse SR optimization can be converted into an equivalent m-sparse quadratic programming, which is non-convex. It then proposed to use a proximal gradient algorithm to obtain a locally optimal solution.

**Strengths:**

- this paper studies an important and practical problem of sparse portfolio optimization, and it is presented with a good clarity.
- The paper proposed to directly optimize the fraction - Sharpe ratio, which can be turned into a nonconvex fractional optimization under constraints. Such an idea to alternatively formulate the problem is natural and very interpretable.
- It is shown theoretically that the proposed proximal gradient algorithm is guaranteed to converge to a locally optimal Sharpe. Experimental results further show that PGA outperforms other baselines in terms of the achieved Sharpe ratios.

**Weaknesses:**

- Although the proposed PGA algorithm is guaranteed to converge to a locally-optimal solution, there is a lack of analysis and results on how suboptimal the converged solution might be in the worst case.
- Moreover this is a lack of discussions on the convergence rate for the proposed PGA algorithm.

**Questions:**

- Directly optimizing the fractional objective can be very unstable with real-world noisy financial data. How does the proposed PGA algorithm compared to the traditional optimization in terms of robustness?
- What are the computational time and costs for the different baselines?

**Limitations:**

Yes.

---

> ### Author Rebuttal · Authors · 2024-08-05
>
> **Answer for Weakness 1**
>
> In fact, under certain conditions, our method can directly converge to a globally optimal solution. Based on Theorem 2 $(i)$ in the original manuscript, we have now further proven more intuitive sufficient conditions for convergence to global optimum. For a detailed proof, please refer to **1. Sufficient conditions for PGA's convergence to a global optimum** in the **'Author Rebuttal for Global Response'**.
>
> Specifically, if one of the following two conditions holds for the limit point $v^*$ of the sequence $\\{v^k\\}$ generated by PGA:
>
> (1) $\\|v^*\\|_0<m$,
>
> (2) $\\|v^*\\|_0=m$ and $\nabla_i f(v^*)>-\epsilon\cdot\min\\{v_i^*|i\in{\rm supp}(v^*)\\}$ for all $i\in\mathbb{N}\backslash{\rm supp}(v^*)$,
>
> then $v^*$ is a globally optimal solution of of model (3.7).
>
> We would like to add that the non-convex fractional model with the non-convex $\ell_0$-constraint presents a highly complex doubly non-convex optimization problem. Despite this complexity, our new result successfully provides theoretically verifiable sufficient conditions for convergence to the global optimum, which is a significant achievement. These rigorous theoretical results have meaningful implications for sparse multi-objective optimization in machine learning.
>
> To further demonstrate that our proposed method can indeed converge to the global optimum of the model, we conducted additional simulation experiments. For more details of the simulation experiments, please refer to **3. Validation of PGA's global optimality through simulation experiments** in the **'Author Rebuttal for Global Response'**. In our simulations with 10,000 random data sets for each of the three different initializations, over 72% of the experiments for each initialization showed that both the normalized error of the iterative sequence (NEIS) $\\|v^k-v^*\\|_2/\\|v^*\\|_2$ and the normalized error of the function value (NEFV) $|f(v^k)-f(v^*)|/|f(v^*)|$ obtained after 500 iterations of PGA were less than $10^{-10}$. We show the plots of NEIS and NEFV in Response Figure 1, and show in Response Table 1 these two normalized errors obtained after 500 iterations of PGA in ten simulation experiments (see the attached PDF file).
>
> In 10,000 experiments, we averaged the normalized errors for cases that did not converge to the global optimum ($>10^{-5}$). The average NEIS values for the three different initializations were 0.5132, 0.5550, and 0.6123, while the average NEFV values were 0.0889, 0.1080, and 0.1334, respectively. This indicates that even when not converging to the global optimum, the local optima achieved by our algorithm exhibit good overall performance.
>
>
> **Answer for Weakness 2:**
>
> Thanks for the insightful comments regarding convergence rate. We have conducted a rigorous theoretical analysis to conclusively demonstrate that the convergence rate of PGA in terms of the function value is $f(v^k)-f(v^*)=O(1/k)$. Since this proof is relatively lengthy, we did not include it here. In the upcoming revised version of the manuscript, we will incorporate the comprehensive proof to substantiate this finding.
>
> **Answer for Question 1:**
>
> We have carefully transformed the fractional model (3.3) into an equivalent quadratic model (3.4) in subtractive form. Hence we just need to solve the quadratic model (3.4), which is more robust than directly solving the fractional model (3.3) with respect to the data noise.
>
>
> **Answer for Question 2:**
>
> The time complexity of S1, S2 and S3 is $O(TN+Nlog(N))$, and the time complexity of PLCT is $O(TN+N^2)$, where $T$ and $N$ denote the window size and the number of assets, respectively. On the other hand, the time complexity of SSMP, SPOLC, SSPO, IPSRM-D, MAXER and mSSRM-PGA is $O(TN\epsilon^{-1})$, where $\epsilon$ denotes the convergence tolerance in the objective function value. Response Table 2 in the attached PDF shows the running times for these methods, which are consistent with their time complexities. Therefore, mSSRM-PGA is competitive in time complexity, and achieves good investing performance.

---

### Official Review · Reviewer_iiAu · 2024-07-16

**Soundness:** 3
**Presentation:** 3
**Contribution:** 2
**Rating:** 6
**Confidence:** 1

**Summary:**

This paper studies Sharpe ratio optimization with sparsity constraints. The paper transforms the original m-sparse fractional optimization problem into an m-sparse quadratic programming problem and develops a proximal-gradient algorithm to solve it. Numerical experiments show that the proposed method improves the Sharpe ratio compared with existing methods.

**Strengths:**

I am not familiar with this area. This paper first represents the Sharpe ratio by substituting the mean and variance with unbiased estimates, and then rewrites the problem in a quadratic objective. From a purely mathematical perspective, this entire procedure is quite standard. What’s interesting seems to be the convergence guarantee and the experiment performance of the proposed algorithm. I have some questions about them but if they turn out justified, I think this work brings fair contribution.

**Weaknesses:**

While the paper transforms the original problem to (3.4), the $\ell_0$ constraint is non-convex. In (3.7), the objective function is still non-convex and it seems to contradict with standard optimization theory that you can guarantee convergence to global optimum in this non-convex objective. Indeed, this is a combinatorial optimization problem and has computation lower-bounds. In sum, I think the theoretical results need further justification on why they do not contradict with existing hardness results.

POST REBUTTAL: the author justified their results in rebuttal.

**Questions:**

See the question in the weakness section.

**Limitations:**

The authors have addressed the limitations.

---

> ### Author Rebuttal · Authors · 2024-08-05
>
> In fact, our method guarantees a locally optimal solution to the non-convex optimization in a general case, which is consistent with standard optimization theory. Only under certain conditions (Theorem 2 $(i)$), the locally optimal solution become a globally optimal solution.
>
> Additionally, based on Theorem 2 $(i)$, we have now further proven more intuitive sufficient conditions for global optimum, during the process of rebuttal. For a detailed proof, please refer to **1. Sufficient conditions for PGA's convergence to a global optimum** in the **'Author Rebuttal for Global Response'**.
>
> Specifically, if one of the following two conditions holds for the limit point $v^*$ of the sequence $\\{v^k\\}$ generated by PGA:
>
> (1) $\\|v^*\\|_0<m$,
>
> (2) $\\|v^*\\|_0=m$ and $\nabla_i f(v^*)>-\epsilon\cdot\min\\{v_i^*|i\in{\rm supp}(v^*)\\}$ for all $i\in\mathbb{N}\backslash{\rm supp}(v^*)$,
>
> then $v^*$ is a globally optimal solution of of model (3.7).
>
> We would like to add that the non-convex fractional model with the non-convex $\ell_0$-constraint presents a highly complex doubly non-convex optimization problem. Despite this complexity, our new result successfully provides theoretically verifiable sufficient conditions for convergence to the global optimum, which is a significant achievement. These rigorous theoretical results have meaningful implications for sparse multi-objective optimization in machine learning.
>
> To further demonstrate that our proposed method has a high probability of converging to the global optimum of the model, we conducted additional simulation experiments. For more details of the simulation experiments, please refer to **3. Validation of PGA's global optimality through simulation experiments** in the **'Author Rebuttal for Global Response'**. In our simulations with 10,000 random data sets for each of the three different initializations, over 72% of the experiments for each initialization showed that both the normalized error of the iterative sequence $\\|v^k-v^*\\|_2/\\|v^*\\|_2$ and the normalized error of the function value $|f(v^k)-f(v^*)|/|f(v^*)|$ obtained after 500 iterations of PGA was less than $10^{-10}$ from the global optimum. We show the plots of $\\|v^k-v^*\\|_2/\\|v^*\\|_2$ and $|f(v^k)-f(v^*)|/|f(v^*)|$ in Response Figure 1, and show in Response Table 1 these two normalized errors obtained after 500 iterations of PGA in ten simulation experiments (see the attached PDF file).

---

> > ### Comment · Reviewer_iiAu · 2024-08-08
> >
> > Thank you for your clarification. I will raise my rating to 6.

---

### Author Rebuttal · Authors · 2024-08-05

We greatly appreciate the reviewers' professional feedback, which has significantly improved this paper. Based on their suggestions, we have added three main components: 1. sufficient conditions for PGA's convergence to a global optimum; 2. analysis of the convergence rate of PGA; 3. validation of PGA's global optimality through simulation experiments. We believe that these components significantly improve the completeness and contribution of our paper.

Next, we provide a detailed description of these three aspects.

**1. Sufficient conditions for PGA's convergence to a global optimum**

We have already proven in Theorem 5 of the original manuscript that the sequence $\\{v^k\\}$ generated by PGA converges to a locally optimal solution of model (3.7).

If one of the following two conditions holds:

$(i)$ $\\|v^*\\|_0<m$,

$(ii)$ $\\|v^*\\|_0=m$ and $\nabla_i f(v^*)>-\epsilon\cdot\min\\{v_i^*|i\in{\rm supp}(v^*)\\}$ for all $i\in\mathbb{N}_N\backslash{\rm supp}(v^*)$,

then $v^*$ is a globally optimal solution of model (3.7).

${\boldsymbol{Proof.}}$ According to item $(i)$ of Theorem 2, to prove the desired result, it suffices to show that

$$
v^*={\rm prox}_{\iota{\tiny\Omega}}\Big(v^*-\frac{1}{\epsilon}\nabla f(v^*)\Big).\ \ \ \ (Eq\ 1)
$$

From the proof of Theorem 5 in Supplementary A.5, we know that

$$
v^*={\rm prox}_{\iota{\tiny\Omega}}(v^*-\alpha\nabla f(v^*))\ \ \ \ (Eq\ 2)
$$

holds. According to the computation of ${\rm prox}_{\iota{\tiny\Omega}}$ in Proposition 3, to guarantee the validity of $(Eq\ 2)$,  we have $\nabla_i f(v^*) = 0$ for all $i\in{\rm supp}(v^*)$. Otherwise, there exists some $i_0\in{\rm supp}(v^*)$ such that $v _ {i{\tiny0}}^*\neq v _ {i{\tiny0}}^*-\alpha\nabla _ {i{\tiny0}} f(v^*)$, which together with Proposition 3 implies that $v^*\neq{\rm prox} _ {\iota{\tiny\Omega}}(v^*-\alpha\nabla f(v^*))$, a contradiction to $(Eq\ 2)$.

Suppose that $\\|v^*\\|_0<m$. Then we have $\nabla_i f(v^*)\geq 0$ for all $i\notin{\rm supp}(v^*)$. Otherwise, there exists some $i_1\in\mathbb{N}\backslash{\rm supp}(v^*)$ such that $v_{i{\tiny1}}^*-\alpha\nabla_{i{\tiny1}}f(v^*)>0$. Note that $\\|v^*\\|_0<m$. The operation of ${\rm prox} _ {\iota{\tiny\Omega}}$ will preserve the positive value $v _ {i{\tiny1}}^* - \alpha\nabla _ {i{\tiny1}}f(v^*)$ instead of truncating it as 0, which violates $(Eq\ 2)$. In this case, we now have $v_i^*-\frac{1}{\epsilon}\nabla_i f(v^*)=v_i^*$ for $i\in{\rm supp}(v^*)$ and $v_i-\frac{1}{\epsilon}\nabla_i f(v^*)\leq0$ for $i\in\mathbb{N}\backslash{\rm supp}(v^*)$, which imply that $(Eq\ 1)$ holds.

Suppose that item $(ii)$ holds. Let $\delta:=\min\\{v_i^*|i\in{\rm{supp}}(\boldsymbol{v}^*)\\}>0$. For $i\in\mathbb{N}\backslash{\rm{supp}}(v^*)$, since $\frac{1}{\epsilon}\nabla_i f(v^*)>-\delta$ and $v_i=0$, we have $v_{i}^*-\frac{1}{\epsilon}\nabla _ i f(v^*)<\delta$. Note that $\|v^*\| _ 0=m$. The operation of ${\rm{prox}} _ {\iota{\tiny\Omega}}$ makes $v_{i}^*-\frac{1}{\epsilon}\nabla _ i f(v^*)=v_i^*$ for $i\in{\rm{supp}}(v^*)$ and $v_{i}^*-\frac{1}{\epsilon}\nabla _ i f(v^*)=0$ for $i\in\mathbb{N}\backslash{\rm{supp}}(v^*)$, that is, $(Eq\ 1)$ holds. This completes the proof.

**2. Analysis of the convergence rate of PGA**

We have conducted a rigorous theoretical analysis to demonstrate that the convergence rate of PGA in terms of the function value is $f(v^k)-f(v^*)=O(1/k)$. Since this proof is relatively lengthy, we did not include it here. In the upcoming revised version of the manuscript, we will incorporate the comprehensive proof to substantiate this finding.

**3. Validation of PGA's global optimality through simulation experiments**

To test validation of PGA's global optimality, we conducted a set simulation experiments by considering the following model:

$$
\min_{v\in\mathbb{R}^N}\left\\{\frac{1}{2}v^\top Q_{\epsilon}v-p^\top v+\iota_{\Omega}(v)\right\\},\ \ \ \ (Model\ 1)
$$

where $Q_{\epsilon}:=Q^\top Q+\epsilon I$ and $\Omega:=\\{v\in\mathbb{R}^N | \ v>0 _ N\ and\ \\|v\\| _ 0\leq m\\}$. The iterative scheme of PGA for solving this model is given by

$$
v^{k+1}={\rm prox} _ {\iota{\tiny\Omega}}\big(v^k-\beta Q_{\epsilon}v^k+\beta p\big),
$$
where $\beta$ is set as $\frac{0.99}{\\|Q\\|_2}$.

In the simulation experiments, we set $\Sigma\in\mathbb{R}^{10\times 10}$ by $\Sigma_{ij}: = 0.5^{|i-j|}$, and use the matlab function 'mvnrnd' to randomly generate a matrix $Q\in\mathbb{R}^{50\times10}$ from the multivariate normal distribution, with mean vector $0_{10}$ and covariance matrix $\Sigma$. We set $p$ as a random vector with components that are randomly generated numbers in the range $[-10,10]$, and casually set $\epsilon=0.001$, $m=3$.

The direct exhaustive approach enumerates all possible support set configurations, totaling $C_{10}^{3}=120$ cases. In each case, we solve a 3-dimension quadratic programming problem. By comparing the optimal solutions corresponding to these 120 cases, we obtain the exact globally optimal solution of $(Model\ 1)$. After that, we can evaluate the optimality of PGA's convergence.

For each experiment, we performed 500 iterations of PGA. To ensure the robustness of our findings, we used three different initializations: $0_N$, $1_N/N$ and $1_N$. We repeated the experiments $10^4$ times for each initialization, with different $Q$ and $p$ in each run. We found that in over 7,200 of the $10^4$ trials, for any of the three initializations, both the normalized error of the iterative sequence $\\|v^k-v^*\\|_2/\\|v^*\\|_2$ and the normalized error of the function value $|f(v^k)-f(v^*)|/|f(v^*)|$ were smaller than $10^{-10}$. Here $v^*$ denotes the globally optimal solution, and $v^k$ represents the iterative sequence at the $k$-th iteration of PGA.

From the simulation experiments, we conclude that the proposed PGA has a high probability (over 72\%) of directly converging to a globally optimal solution of $(Model\ 1)$. This finding is consistent with our newly proven sufficient conditions for global optimality.

---

### Decision · Program_Chairs · 2024-09-25

**Decision:**

Accept (poster)

**Comment:**

I would like to thank both the authors and reviewers for the good discussion. The reviewers agree that the paper is interesting and all recommend acceptance although there is no single strong supporter. Having looked at the paper and read the reviews, I recommend acceptance (but really borderline acceptance), however I urge the authors to seriously take the reviewers feedback into account when preparing the camera-ready version.